# Fused in sarcoma (FUS) inhibits milk production efficiency in mammals

Haili Shao[1,2,6], Jipeng Huang[1,2,6], Hui Wang [1,6], Guolei Wang[3,6], Xu Yang [1,2], Mei Cheng[1,2], Changjie Sun[1,2], Li Zou[1,2], Qin Yang[1], Dandan Zhang[4], Zhen Liu[1], Xuelong Jiang[1], Lei Shi [1], Peng Shi [1,5], Baowei Han[4] ✉ & Baowei Jiao [1,5] ✉

Efficient milk production in mammals confers evolutionary advantages by facilitating the transmission of energy from mother to offspring. However, the regulatory mechanism responsible for the gradual establishment of milk production efficiency in mammals, from marsupials to eutherians, remains elusive. Here, we find that mammary gland of the marsupial sugar glider contained milk components during adolescence, and that mammary gland development is less dynamically cyclic compared to that in placental mammals. Furthermore, fused in sarcoma (FUS) is found to be partially responsible for this establishment of low efficiency. In mouse model, FUS inhibit mammary epithelial cell differentiation through the cyclin-dependent kinase inhibitor p57Kip2, leading to lactation failure and pup starvation. Clinically, FUS levels are negatively correlated with milk production in lactating women. Overall, our results shed light on FUS as a negative regulator of milk production, providing a potential mechanism for the establishment of milk production from marsupial to eutherian mammals.

The mammary gland, a defining feature of mammals, plays a crucial role in facilitating the efficient delivery of nutrition from mother to offspring through lactation[1]. Throughout mammalian evolution, there has been a gradual increase in delivery efficiency from marsupials to eutherians[2,3]. Although both marsupials (e.g., tammar wallabies and sugar gliders) and eutherians (e.g., mice) produce milk, they exhibit distinct differences in lactation strategies. Marsupials, characterized by a short gestation period with limited energy resources available for the developing fetus, rely on a prolonged suckling period in the mother's pouch for nourishment and growth, lasting ~300 days in the tammar wallaby (*Macropus eugenii*)[4] and 70–90 days in the sugar glider (*Petaurus breviceps*). In contrast, rats of comparable weight to sugar gliders exhibit a considerably shorter lactation period of only 21 days[5]. The efficient energy transfer gained during mammary gland evolution in rats allows their young to complete development within a relatively short period, and then survive independently.

In humans, breastfeeding reduces the risk of many diseases in mothers, including breast cancer, ovarian cancer, postpartum depression, and cardiometabolic disorders[6,7]. In addition, breastfeeding protects infants from gastrointestinal infections, adaptive dermatitis, respiratory diseases, and obesity[8]. However, less than 25% of infants are exclusively breast-fed[9], with insufficient maternal milk secretion being an important reason[10]. Therefore, elucidating the precise regulatory mechanism underlying lactation during evolution is critical for understanding the etiology of postpartum hypogalactia.

Mammary epithelial cells (MECs) increase exponentially during pregnancy and differentiate into alveolar cells in late pregnancy, when MEC proliferation rapidly declines as the glands devote themselves to

[1]National Key Laboratory of Genetic Evolution & Animal Models, Kunming Institute of Zoology, Chinese Academy of Sciences, Kunming, Yunnan 650201, China. [2]Kunming College of Life Science, University of the Chinese Academy of Sciences, Kunming, Yunnan 650201, China. [3]Department of Obstetrics, Weifang People's Hospital, Weifang, Shandong 261042, China. [4]Luoyang Maternal and Child Health Hospital, Luoyang, Henan 471000, China. [5]KIZ-CUHK Joint Laboratory of Bioresources and Molecular Research in Common Diseases, Kunming Institute of Zoology, Chinese Academy of Sciences, Kunming, Yunnan 650203, China. [6]These authors contributed equally: Haili Shao, Jipeng Huang, Hui Wang, Guolei Wang. ✉e-mail: hbw315@126.com; jiaobaowei@mail.kiz.ac.cn

differentiation[11–13]. Through ductal branching and extension, the development and remodeling of the mammary duct, crucial for lactation, rely on coordinated MEC proliferation and the regulation and maintenance of cell differentiation[14]. However, the specific factors driving MEC differentiation and mammary gland development remain unclear. Previous studies have suggested that post-transcriptional regulation may play a critical role in adaptive divergence and evolution[15–17]. RNA-binding proteins (RBPs) are key players in the post-transcriptional regulation of gene expression[18–20].

In this work, we discover distinct differences regarding mammary gland development in marsupial sugar gliders and eutherian mice. Notably, unlike the dynamic cycle observed in mice, sugar gliders exhibit consistent morphological characteristics throughout the different stages of development. Furthermore, we identify Fused in sarcoma (FUS) as a key regulatory protein for the above lactation efficiency. Overall, our study reveals the pivotal role of FUS as a regulator of lactation, providing valuable insights into the underlying mechanisms governing milk production processes.

## Results

### Sugar gliders exhibit lower milk production control ability

Our current understanding of mammary gland development in marsupials is limited. To investigate the differences in mammary glands between marsupials and eutherian mammals, we explored mammary gland development in sugar gliders compared to that in mice. We first conducted a detailed morphological analysis of the mammary glands of female sugar gliders at four stages (virgin, pregnancy, lactation, and involution) (Fig. 1a). In contrast to mice, which demonstrated a dynamic presence of milk-producing alveolar units, the sugar gliders featured evident mammary gland alveolar structures not only during the lactation period but also during the virgin and pregnancy stages (Fig.1b). Furthermore, the number of mammary gland ducts in the sugar gliders, both during the virgin and lactation stages, was significantly lower compared to that in mice (Fig. 1c, d), as demonstrated by immunofluorescence staining of the ductal marker CD133[21–23] (Supplementary Fig. 1a, b). Interestingly, milk-like secretions were observed in the mammary gland of sugar gliders during the virgin stage, whereas mice exhibited milk secretion exclusively during the lactation period (Fig. 1e). To ascertain whether the secretions observed in virgin mammary glands of sugar gliders were indeed milk, we conducted a comprehensive composition analysis. Milk is known to contain essential nutritional components such as proteins, fats, and lactose[24]. Immunofluorescence staining revealed that mice exclusively produced milk protein in their mammary glands during lactation, while, in sugar gliders, milk protein was also detected in the virgin stage of mammary gland development (Fig. 1f). Oil Red staining further demonstrated the presence of mammary gland milk fat in both the virgin and lactation stages in sugar gliders, while this was only present during the lactation period in mice (Fig. 1g). Furthermore, we used Raman spectroscopy[25,26] to detect the Raman spectra of milk from both mice and sugar gliders, as well as the mammary gland contents during the virgin and lactation. Only the constituents from the lactating mouse mammary glands corresponded precisely with the milk spectra, there were significantly different peak shape and intensity variations in the Raman spectra of mammary contents from virgin compared to mice milk spectra. Conversely, for sugar gliders, the Raman spectra of mammary gland contents during lactation matched perfectly with sugar glider milk spectra, and similarly, the spectra from virgin mammary gland contents also aligned completely with milk spectra (Fig. 1h). These findings confirmed that sugar gliders were capable of secreting milk components during non-lactating periods. Unlike in mice, the lack of precise regulation of milk secretion in sugar gliders, which occurs even when offspring do not require milk, suggests an inefficient utilization of energy by the parent. Additionally, sugar gliders contained significantly fewer mammary ducts than mice.

Overall, these results indicate that sugar gliders exhibit a lower regulatory capacity for milk production compared to mice.

### FUS is a potential regulator of lactation efficiency in mammals

The regulation of lactation in eutherian mammals follows a dynamic cyclic pattern, with milk components being synthesized during gestation and secreted during lactation[27]. In contrast, our results showed that lactation regulation in sugar gliders remained unchanged, with milk components present at various stages of mammary gland development. Based on these disparities in lactation strategies between eutherians and marsupials, we hypothesized that certain key factors involved in the regulation of lactation may remain unchanged throughout marsupial mammary gland development. To explore these key factors and their evolutionary significance, we aimed to identify genes that were differentially expressed in eutherian mammals but did not show significant expression changes during marsupial mammary gland development. We first collected mammary gland samples from sugar gliders and mice during the virgin and lactation stages for RNA-seq analysis. By comparing the gene expression profiles, we identified genes that displayed no significant changes ($P > 0.05$; |fold-change|<2) between the two stages in sugar glider mammary glands, while exhibiting significant changes ($P < 0.05$; |fold-change|> 2) between the two stages in mouse mammary glands, yielding a total of 2214 genes (Supplementary Data 1). To further refine this gene list, we incorporated differentially expressed genes (DEGs) between lactating and non-lactating mammary glands in well-recognized dynamic species, including cattle (GSE116079), pigs (GSE30704), and mice (GSE12247)[28]. Among the DEGs across the three species, a total of 26 genes exhibited significant fluctuations ($P < 0.05$) in expression levels between lactation and virginity (Supplementary Table 1). In combination with the above two gene lists, five genes demonstrated significant changes in mice but not in sugar gliders during both stages (Fig. 2a). Notably, among these genes, *Fus* displayed the smallest fold-change (−1.02) and the least significant level of change ($P = 0.87$) in the sugar glider mammary gland (Supplementary Data 2).

FUS expression in the sugar glider mammary gland remained largely stable over time. This was consistent at the protein level, as shown by immunohistochemical staining (Fig. 2b) and western blotting (Supplementary Fig. 1c), and at the mRNA level, as indicated by quantitative real-time polymerase chain reaction (RT-qPCR) results (Fig. 2c). In contrast, eutherian mice exhibited a significant reduction in FUS protein levels during late pregnancy and lactation compared to levels observed during the virgin and involution stages (Fig. 2b). These findings were confirmed at the protein level by western blot analysis (Supplementary Fig. 1d, e) and immunofluorescence staining (Supplementary Fig. 1f) and at the mRNA level by RT-qPCR (Supplementary Fig. 1g). Furthermore, low FUS expression during lactation was also evident in eutherians beyond mice, including horseshoe bats (*Rhinolophus sinicus*) (Fig. 2b), suggesting a common pattern across eutherian species. Taken together, these findings suggest that the identified FUS protein potentially serves as a regulator of lactation efficiency in mammals.

### FUS expression levels are crucial for lactation efficiency in mammals

We next explored the reasons behind the distinct expression patterns of FUS between marsupials and eutherians. As genetic diversity is a key parameter for understanding the evolutionary process[29], we compared the amino acid sequences of FUS between eutherians and marsupials. Our findings revealed that a glycine and serine enrichment sequence (GS-sequence, PGS: GGGGGGGGSG for sugar glider, *Petaurus breviceps*, MGS: GGGGSGSGGG for tammar wallaby, *Macropus eugenii*, and SGS: GSSSGG for tasmanian devil, *Sarcophilus harrisii*) was absent in the glycine-rich domain (GR) of the FUS gene coding region from species with a strong lactation capability (Fig. 2d

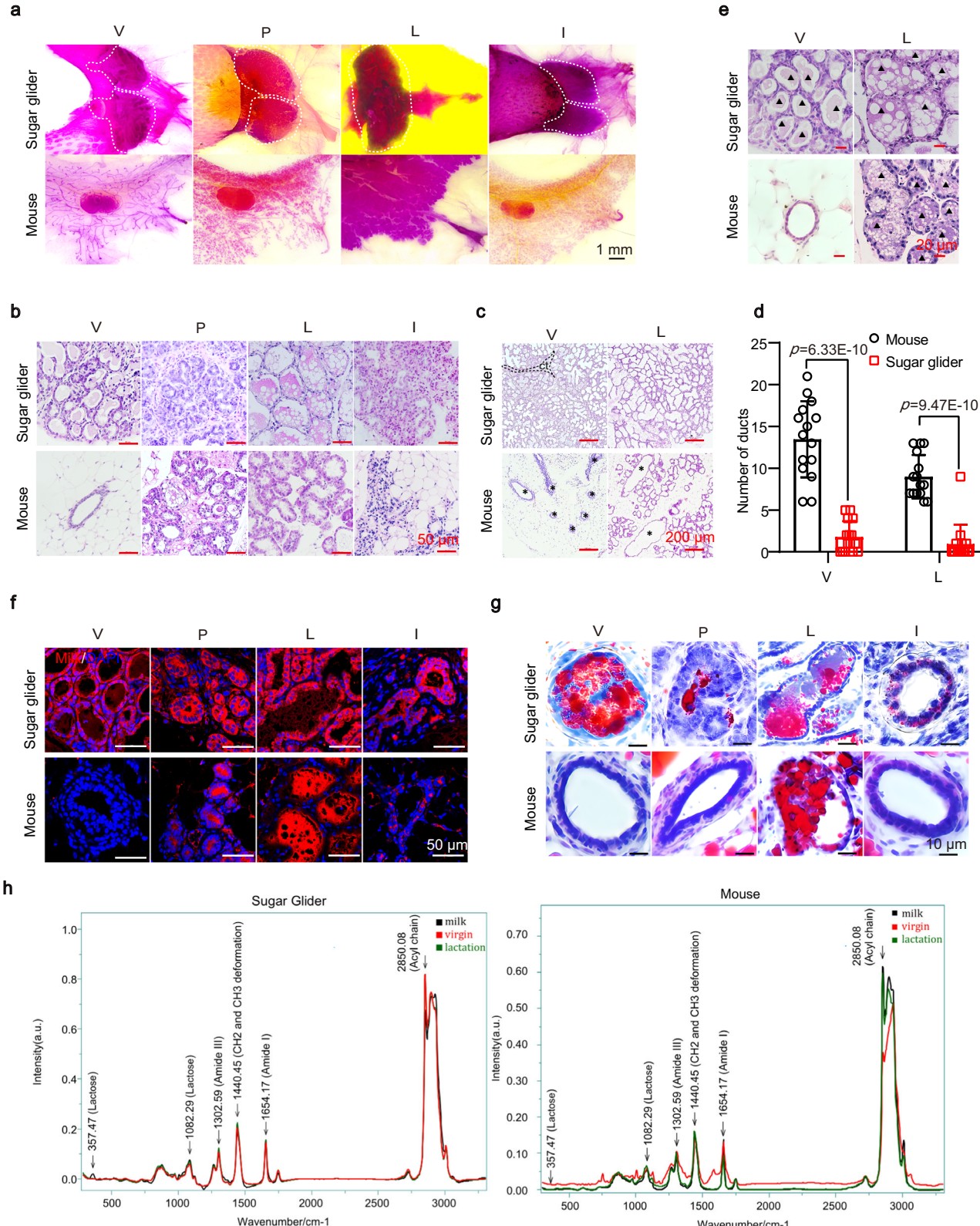

and Supplementary Fig. 2). These results imply that the loss of the GS-sequence in the GR decreases the FUS protein level during the lactation stage in those species with strong lactation ability. Conversely, in the sugar glider, this sequence was retained, and the protein level of FUS remained stable over time in the mammary gland stages. To determine the impact of the GS-sequence on FUS protein levels, the GS-sequence from marsupials was inserted into the GR

region of human (hFUS) and mouse (mFUS) FUS sequences, named as mFUS-PGS, mFUS-MGS, mFUS-SGS, hFUS-PGS, hFUS-MGS, and hFUS-SGS. These constructs were then expressed in various cell lines (Fig. 2e). Of note, the presence of inserts significantly increased both the mRNA and protein levels of FUS (Fig. 2f, g and Supplementary Fig. 1h, i). The longer inserts (PGS and MGS) induced much more stable FUS levels compared to the shorter insert (SGS), further

**Fig. 1 | Sugar glider mammary gland produces milk during adolescence.**
**a** Whole-mount carmine staining of mammary glands (white curves) in sugar gliders and mice at virgin (V, 2-month-old mice and 8-month-old sugar gliders, $n = 3$), pregnancy (P, mice and sugar gliders in second trimester of first pregnancy, $n = 4$), lactation (L, mice and sugar gliders at mid-lactation during first pregnancy, $n = 3$), and involution (I, mice weaned at 2 days and sugar gliders weaned at 16 days, $n = 3$) stages. The experiments were performed to have three biological replicates independently with similar results. Scale bar: 1 mm. **b**–**e** H&E staining of mammary glands (**b**), ducts (**c**), and milk-like secretions (**e**) in mammary glands of sugar gliders and mice at various developmental stages. **d** Statistical analysis of the number of mammary gland ducts in sugar gliders and mice during virgin and lactation stages, based on the quadruple field of view. V (mouse: $n = 15$; sugar glider: $n = 15$); L (mouse: $n = 15$; sugar glider: $n = 15$). ct connective tissue. Data were

means ± SD. An unpaired $t$-test was used to evaluate statistical significance. $P$ values are indicated in the chart. **f** Immunofluorescence staining of milk (red) and DAPI (blue) in mammary gland sections of sugar gliders and mice at virgin, pregnancy, lactation, and involution stages. Scale bar: 50 μm. The experiments were performed to have three biological replicates independently with similar results. **g** Oil Red staining of lipids (red) in mammary gland sections of sugar gliders and mice at virgin, pregnancy, lactation, and involution stages. Scale bar: 10 μm. The experiments were performed to have three biological replicates independently with similar results. **h** Raman spectroscopy and characteristic peaks of milk secretion and mammary gland contents during virgin and lactation stages in sugar gliders and mice. Raman shift: 1302 cm$^{-1}$ (Amide III); 1440 cm$^{-1}$ (CH2 and CH3 deformation vibrations); 1654 cm$^{-1}$ (Amide I); 2850 cm$^{-1}$ (stretching vibrations of CH, NH, and OH groups). Source data are provided as a Source Data file.

indicating that GS-enriched sequences are required for maintaining FUS expression. Furthermore, we investigated whether the GS-sequence had an effect on the mRNA stability of *Fus* through mRNA stability experiments. The results showed that the insertion of GS-sequence promoted the mRNA stability of *Fus* (Fig. 2h and Supplementary Fig. 1j) and FUS protein levels (Supplementary Fig. 1k). Thus, these results suggest that FUS expression is regulated by GS-sequence across different species.

### High FUS expression in MECs results in lactation failure

To validate the role of FUS in lactation efficiency, we recovered FUS expression during the late pregnancy and lactation stages of the mammary gland using a transgenic mouse model overexpressing FUS. FUS expression was enhanced in *Fus* floxed mice (*Fus*$^{fl/fl}$) crossed with transgenic mice expressing Cre recombinase (Cre) under the control of the whey acidic protein (WAP) promoter, specifically targeting luminal epithelial cells from the mid-pregnancy to lactation stages[30] (Supplementary Fig. 3a). The efficiency of the *Fus* overexpression system in mice was validated through various assays (Fig. 3a, b and Supplementary Fig. 3b, c). In *Fus*$^{fl/wt}$/WAP-Cre (*Fus*-OE) mice, FUS predominantly localized in the nucleus, as revealed by immunofluorescence staining (Supplementary Fig. 3d). We conducted further investigations to verify the specificity of Cre recombinase expression driven by the WAP promoter. This involved isolating and analyzing different brain regions and the spinal cord for the presence of FUS and Cre recombinase using western blotting. The analysis confirmed that Cre recombinase was not present in these samples, indicating no unintended Cre activity in the cerebral cortex, limbic system, or spinal cord (Supplementary Fig. 3e, f). Additionally, in the *Fus*-OE group, the FUS levels were comparable to those in the WAP-Cre group within the brain and spinal cord, suggesting no overexpression of FUS in these areas (Supplementary Fig. 3e, f). GFP expression, exclusive to *Fus*-OE mice, was also detected (Supplementary Fig. 3c), with no GFP presence in the samples from the WAP-Cre and *Fus*$^{fl/fl}$ groups. These findings confirm the successful establishment of a *Fus* gene overexpression mouse model.

To assess the effects of FUS on offspring survival, we initially analyzed the survival rates of pups from mixed litter sizes. Notably, we observed significantly lower survival rates among first-litter pups born to *Fus*-OE mice compared to pups born to WAP-Cre mice. Most pups in the *Fus*-OE group did not survive beyond lactation day 2 (L2) (Supplementary Fig. 3g), with the decreased survival rates stabilizing after the fourth day (Fig. 3c). Next, to eliminate any variations in nutrition caused by different litter sizes, we calculated the pup survival rates for the same-sized litters (6, 7, or 8 pups per litter), which showed a similar pattern (Fig. 3d and Supplementary Fig. 3g–i). Furthermore, despite improvements in pup survival in the overexpression group during the second lactation, a similar trend of declining pup survival was observed (Fig. 3e and Supplementary Fig. 3j).

To gain a better understanding of the role of FUS, we closely monitored the weight of surviving pups by adjusting the litter size to six at the end of L2. When comparing the average weight and weight gain of surviving pups born to *Fus*-OE mothers during their first (Fig. 3f, black curves) and second lactations (Fig. 3g and Supplementary Fig. 3k), the significant decreases in weight and weight gain were observed in comparison to pups born to WAP-Cre mothers. To exclude possible defects in pups, we conducted cross-fostering experiments, whereby pups born to *Fus*-OE and WAP-Cre mothers were switched and nursed by WAP-Cre and *Fus*-OE mothers, respectively, starting at either L2 or L10. The results demonstrated that pups born to *Fus*-OE mothers but fostered by WAP-Cre mice from L2 onwards exhibited normal body weights and survival rates (Fig. 3f, red curves). In addition, the lower average weight of surviving pups born to *Fus*-OE mothers could be rescued when fostered by WAP-Cre mothers starting at L10 (Fig. 3f, blue curves). Thus, these findings suggest that lower FUS expression is required for offspring survival in mice.

### High FUS expression induces insufficient milk production

To identify factors contributing to the mortality of pups nursed by *Fus*-OE mice, we next focused on milk, the primary source of energy for offspring survival. The stomachs of *Fus*-OE contained minimal milk (Fig. 4a). To exclude the possibility of muscle weakness affecting suckling ability, we conducted cross-fostering experiments. For the pups given birth by *Fus*-OE mothers, 1/3 of the pups were fed by the foster mothers of WAP-Cre genotype or *Fus*$^{fl/fl}$ genotype on day 1, respectively. The breastfeeding status was observed after 8 h of cross-fostering. Results demonstrated that the stomachs of pups born to *Fus*-OE mothers but fostered by WAP-Cre mice or *Fus*$^{fl/fl}$ mice contained sufficient milk, implying that pups born to *Fus*-OE mothers had normal suckling abilities (Supplementary Fig. 4a). Conversely, control, pups born to WAP-Cre mothers but fostered by *Fus*-OE mice contained minimal milk in their stomachs (Supplementary Fig. 4b). These results suggest that the pups of both *Fus*-OE and WAP-Cre genotypes could suckle milk normally, thereby excluding impaired suckling as a contributing factor to their mortality. We collected milk from lactating mice using established protocols for volume analysis, as described in our previous study[31]. Notably, compared with WAP-Cre mice, *Fus* overexpression mice showed a significant decrease in milk volume at both L2 and L10 (Fig. 4b and Supplementary Fig. 4c). Immunofluorescent and immunohistochemical analyses further confirmed that *Fus* overexpression inhibited milk production (Fig. 4c and Supplementary Fig. 4d, e). Furthermore, the expression levels of the α-casein, β-casein, and *Wap* genes, which encode milk protein, as well as the *Prlr* gene, a key regulator of lactation[32–34], were significantly downregulated in the MECs of the *Fus*-OE group (Fig. 4d).

### High FUS expression results in sparse alveolar structures

We next examined whether alveolar dysplasia occurred in *Fus*-OE mice using whole-mount and hematoxylin-eosin (H&E) staining. Whole-

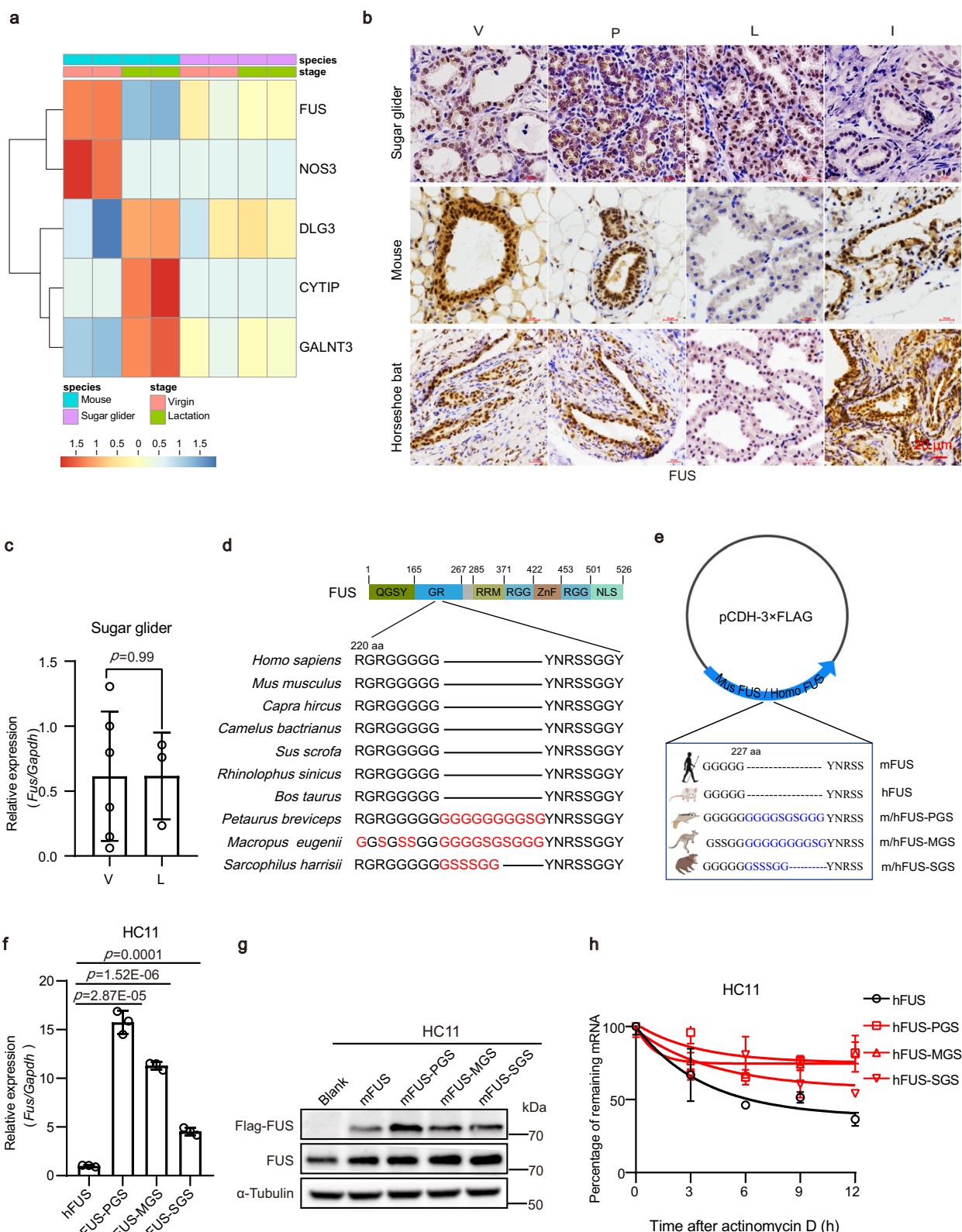

mount analysis revealed notable atrophy and sparsity of secretory lobuloalveolar structures in *Fus*-OE mammary glands, even immediately after delivery, with the sparse alveolar pattern persisting until L20 (Fig. 4e). H&E staining further confirmed the sparse distribution of alveolar structures in the *Fus*-OE group (Fig. 4f). In addition to the WAP-Cre system, we verified these phenotypes in another *Fus* overexpression system driven by K14 under tamoxifen induction

(Supplementary Fig. 3a). Whole-mount and H&E staining again demonstrated the presence of sparse alveolar structures in the *Fus*-OE group (Supplementary Fig. 4f, g). Upon closer examination at higher magnification, all observed alveolar structures appeared intact without signs of collapse or premature degeneration (Fig. 4f and Supplementary Fig. 4g), suggesting that the sparse alveolar phenotype in the *Fus*-OE group was not attributed to involution.

**Fig. 2 | FUS is a potential inhibitor of lactation in mammals. a** Expression profiles of five genes showing significant differences between virgin and lactation stages in mammary glands of three eutherian species, while exhibiting no significant changes in mammary glands of sugar gliders. Heatmap colors represent gene expression levels. Red to dark blue gradient corresponds to decreased gene expression level. **b** Immunohistochemical staining of FUS in mammary glands obtained from sugar gliders, C57BL/6 mice, and horseshoe bats at different developmental stages (V, virgin; P, pregnancy; L, lactation; I, involution), with three sugar gliders, five mice, and three horseshoe bats for each stage. The experiments were performed to have three biological replicates independently with similar results. Scale bar: 20 μm. **c** Relative mRNA expression of *Fus* in sugar gliders at virgin (V, n = 6) and lactation (L, n = 3) stages. Data were means ± SD. An unpaired *t*-test was used to evaluate statistical significance. **d** Sequence alignment of FUS amino acid in 10 mammalian species. Red represents non-conserved regions. Details are provided in Supplementary Fig. 2. **e** Full-length (FL) and mutant (GS-sequence inserted) human or

mouse FUS was cloned into pCDH-CMV-MCS-EF1-Puro lentiviral expression vector, i.e., mFUS (pCDH-3 × FLAG-Mus FUS), mFUS-PGS (pCDH-3 × FLAG-Mus FUS-PGS, inserted GS-sequence of *Petaurus breviceps*), mFUS-MGS (pCDH-3 × FLAG-Mus FUS-MGS, inserted GS-sequence of *Macropus eugenii*), mFUS-SGS (pCDH-3 × FLAG-Mus FUS-SGS, inserted GS-sequence of *Sarcophilus harrisii*), hFUS (pCDH-3 × FLAG-Homo FUS), hFUS-PGS, hFUS-MGS, and hFUS-SGS, respectively, then homologously or heterologously expressed in HC11 and MCF10A cell lines. The human and animal figures were created with BioRender.com. **f, g** Relative mRNA expression of *Fus* (**f**) and western blot analysis of FUS protein levels (**g**) in HC11 cells (mouse MEC line) transfected with specified vector. **h** qRT-PCR analysis of *Fus* mRNA expression in HC11 cells treated with actinomycin D after overexpression of FUS full-length (hFUS), and FUS mutant (hFUS-PGS, hFUS-MGS, and hFUS-SGS). Data were means ± SD of three independent experiments in graphs (**f, h**). An unpaired *t*-test was used to evaluate statistical significance. *P* values are indicated in the chart (**f**). Source data are provided as a Source Data file.

Collectively, these data demonstrate that increased FUS expression in the mammary gland can lead to a sparse distribution of alveolar structures.

## FUS inhibits MEC differentiation

Upon pregnancy, MECs undergo differentiation into alveolar cells, which subsequently form lactation lobules responsible for milk secretion[12]. Thus, we next tested whether the sparse alveolar structures were the result of reduced MEC differentiation. *Stat5, Elf5, and Nfib*, markers of MEC differentiation, play crucial roles in regulating the expression of milk proteins and other differentiation-related proteins involved in secretion during alveolar differentiation[35,36]. Our results showed that the expression levels of the three markers were significantly decreased in the *Fus*-OE mammary gland at L0 (Fig. 5a). Immunostaining results confirmed the absence of these marker proteins in the *Fus*-OE group (Fig. 5b and Supplementary Fig. 5a), indicating that FUS inhibits MEC differentiation.

To further investigate the impact of FUS on MEC differentiation, we utilized the well-established MEC differentiation model in vitro. HC11 is a normal immortalized PRL-positive MECs derived from the mammary tissue of pregnant BALB/C mice[37]. HC11 cells can be induced to differentiate into alveolar-like cells in the presence of prolactin and other factors, closely mimicking the process of milk production in vivo (Fig. 5c). Alveolar-like cells can form milk-like dome-shaped structures, which can serve as indicators of MEC differentiation[38]. Therefore, we performed *Fus* knockdown in HC11 cells using short hairpin RNA (shRNA) (Supplementary Fig. 5b), then evaluated dome formation. Results showed that *Fus* knockdown significantly promoted dome formation (Fig. 5d, e), while overexpressing *Fus* in HC11 cells resulted in reduced dome formation (Supplementary Fig. 5c, d). Dome formation assays were also carried out in primary mammary cells, revealing that *Fus* overexpression led to a reduction in dome formation (Fig. 5f, g), whereas *Fus* knockdown led to a significant increase in dome formation (Supplementary Fig. 5e, f). We analyzed the expression of milk-related genes, including *Wap, α-Casein, β-Casein*, and *Prlr*. These genes showed significant up-regulation after treatment with prolactin in the *Fus* knockdown group, as indicated by RT-qPCR analysis (Supplementary Fig. 5g). These findings are consistent with previous research showing that HC11 cells can synthesize milk proteins when stimulated by lactogenic hormones, differentiate into alveolar-like cells, and form dome-shaped structures[38,39]. Furthermore, the expression of *Fus* was significantly reduced upon HC11 differentiation (Fig. 5h, i). These findings provide additional evidence supporting the inhibitory role of *Fus* in MEC differentiation.

## FUS regulates coordination between cell cycle exit and MEC proliferation

To further explore the mechanism underlying the impact of *Fus* on milk secretion disorders and MEC differentiation, we extracted RNA

from the breast tissue of *Fus*-OE and WAP-Cre mice at L1 and performed transcriptome sequencing. Based on the Kyoto Encyclopedia of Genes and Genomes (KEGG) analysis of the transcriptome data, we found that DEGs between the *Fus*-OE and WAP-Cre groups were primarily associated with cell cycle and DNA replication (Fig. 6a). Cellular division is essential for generating appropriate numbers of cells, but subsequent exit from the cell cycle is necessary to permit differentiation[40,41]. The duration of the G1 phase is pivotal in establishing a window of opportunity for DNA replication, pluripotency exit, and differentiation commencement[42]. To determine whether FUS regulates MEC differentiation by controlling the cell cycle, we detected the cell cycle in HC11 cells following *Fus* knockdown. Flow cytometry revealed that cell cycle progression was arrested in the G0/G1 phase upon *Fus* knockdown (Fig. 6b). Furthermore, the expression of negative cell-cycle regulators (P21) was up-regulated, and the expression of positive cell-cycle regulators (p-RB, Cyclin D1 and Cyclin E1) was downregulated after *Fus* knockout, resulting in cell cycle arrest in the G1 phase (Fig. 6c). These results suggest that proper cell-cycle exit and MEC differentiation rely on the appropriate expression level of FUS. In addition, the clone formation assay demonstrated that *Fus* knockdown in HC11 and primary cells impaired clone formation ability (Fig. 6d and Supplementary Fig. 5h, i).

As proliferation and differentiation exhibit an inverse relationship during development[42], we further detected the effects of FUS on proliferation. Immunofluorescence analysis using the proliferation marker Ki67 revealed a significantly higher number of proliferating MECs in *Fus*-OE mice compared to WAP-Cre mice at the lactating stage (Supplementary Fig. 5j), as confirmed by immunohistochemical staining and analysis (Fig. 6e, f and Supplementary Fig. 5k, l). To substantiate the regulatory effects of FUS in vitro, we assessed cell proliferation using the 3-(4,5-dimethylthiazol-2-yl)−5-(3-carbox-ymethoxyphenyl)−2-(4-sulfophenyl)−2H-tetrazolium (MTS) assay in HC11 and primary mammary gland cells. Results showed that cell proliferation was significantly reduced after *Fus* knockdown in both cell types (Fig. 6g and Supplementary Fig. 5m). In addition, proliferating cell nuclear antigen (PCNA), a cell proliferation marker, was also downregulated during differentiation (Fig. 5i).

Taken together, these findings indicate that high FUS expression inhibits MEC differentiation by inhibiting cell cycle exit.

## FUS binds to *p57Kip2* mRNA and decreases its stability

To further elucidate the mechanism by which FUS regulates the cell cycle, we investigated its RNA-binding properties. As a known RNA/DNA-binding protein, FUS binds to and interacts with RNA in a variety of ways, thereby modulating downstream signaling[43]. Using online tools (http://www.tartaglialab.com/), we predicted the potential binding between FUS and mRNAs related to the cyclin-dependent kinase (CDK) inhibitor (CKI) (Supplementary Data 3), consistent with our observations that *Fus* inhibited cell

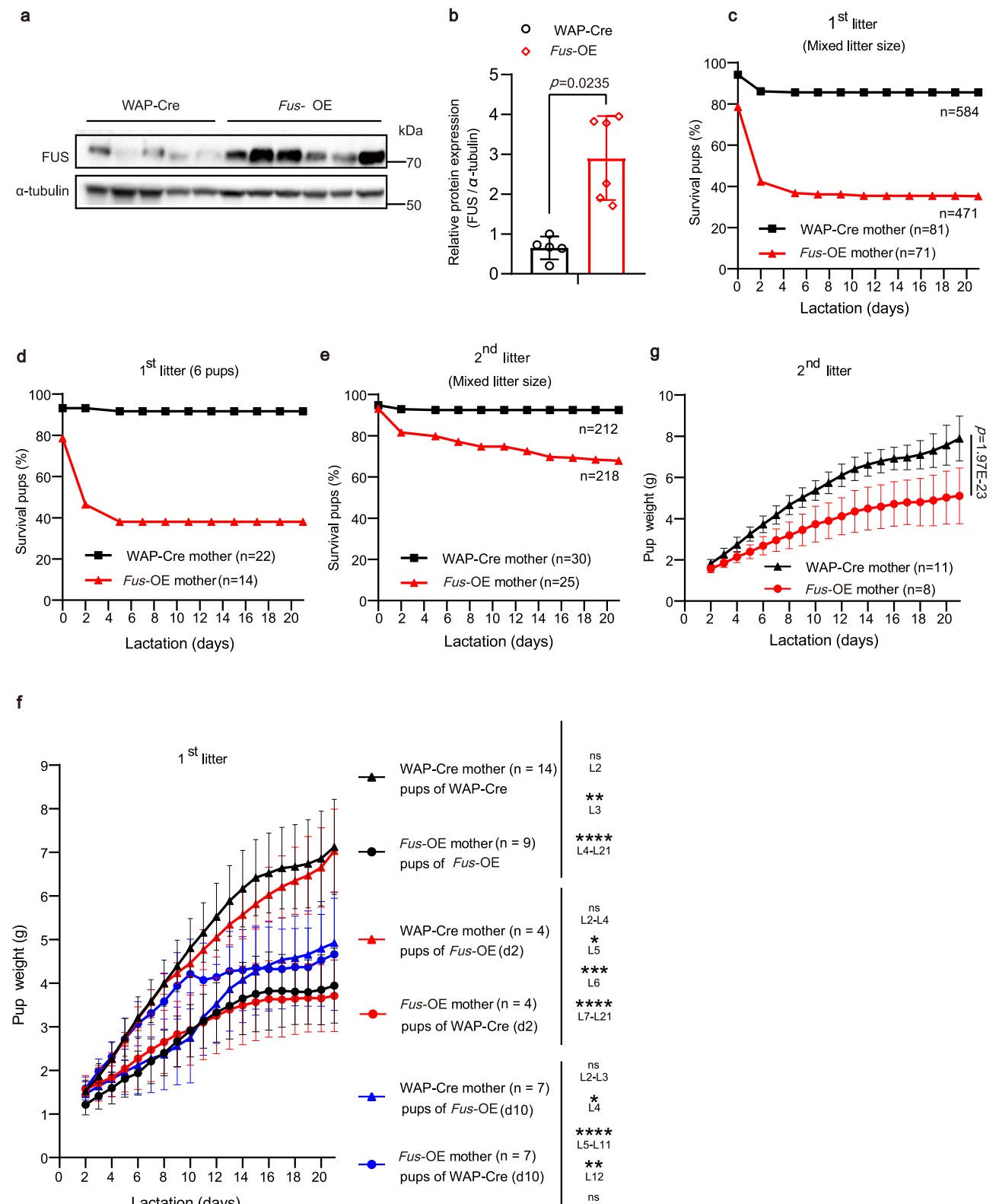

differentiation and promoted cell proliferation. We selected p57Kip2, which had a high ranking in terms of combination ability, for the follow-up study. Notably, FUS binds to the 3'UTR of *p57Kip2*, as confirmed by RNA pull-down assay (Fig. 6h and Supplementary Fig. 6a) and RNA immunoprecipitation (RIP) assay (Fig. 6i and Supplementary Fig. 6b). The p57Kip2 protein is a Cip/Kip family member of CKIs, encoded by the *Cdkn1c* gene, and plays an important role in

the regulation of cell proliferation and differentiation during mammalian development[44].

RNA-binding proteins can regulate mRNA stability through interaction with the 3'UTR of mRNA[45]. To further investigate whether FUS affects the stability of *p57Kip2* by binding to its 3'UTR, we examined the decay of *p57Kip2* mRNA following treatment in transfected HC11 cells with the RNA Pol II inhibitor actinomycin D to stop

**Fig. 3 | FUS overexpression induces failure of both pup viability and growth.**
**a, b** Western blot (**a**) and relative quantification (**b**) of FUS protein levels in mammary glands from *Fus*-OE and WAP-Cre mice at L0. Western blotting: *Fus*-OE mice, $n = 6$; WAP-Cre mice, $n = 5$. RT-qPCR: *Fus*-OE mice, $n = 7$; WAP-Cre mice, $n = 7$. **c**–**e** Pup survival rates from WAP-Cre and *Fus*-OE mothers during lactation stage. Overall pup survival rates in different-sized litters in first (**c**) and second (**e**) lactations. **d** Pup survival rates from WAP-Cre and *Fus*-OE mothers with six pups per litter in the first lactation. **f** Body weights of surviving pups nursed by *Fus*-OE versus WAP-Cre mothers. Body weights of surviving pups nursed by biological mother (*Fus*-OE or WAP-Cre, black lines) and cross-nursed by foster mother (WAP-Cre or *Fus*-OE) at L2 (red lines) and L10 (blue lines), recorded at the beginning at L2. The litter size was adjusted to six for each foster mother on L2. **g** Body weights of pups after litter size adjustment to seven on L2 during the second lactation. Data were means ± SD. An unpaired *t*-test was used to evaluate statistical significance. The statistical test used was two-sided in Fig. 3b, f, g. *P* values are indicated in the chart (**b**, **g**). The exact *P* values of the graph **f** are listed in the Source Data file. *$P < 0.05$; **$P < 0.01$; ***$P < 0.001$; ****$P < 0.0001$; ns no significant difference. Source data are provided as a Source Data file.

transcription. Results revealed a significant increase in *p57Kip2* mRNA levels in the absence of *Fus* (Fig. 6j), suggesting that the loss of FUS promoted *p57Kip2* mRNA stability. To further determine whether FUS regulates *p57Kip2* mRNA stability through the 3′UTR, the 3′UTR of *p57Kip2* mRNA was inserted into downstream of the green fluorescent protein (GFP) gene in a mammalian expressive vector (Fig. 6k). Changes in GFP expression were then assessed upon *Fus* knockdown, serving as an indicator of FUS-mediated regulation of *p57Kip2* via its binding to the 3′UTR. As expected, *Fus* knockdown using shRNA resulted in significantly higher GFP expression in the *GFP-p57Kip2-3′ UTR* group compared with the scramble control (Fig. 6l). Conversely, *Fus* overexpression reduced GFP expression (Fig. 6m). Furthermore, to confirm the regulatory effects of FUS on p57Kip2 in vitro, knockdown of *Fus* in HC11 cells led to a substantial increase in p57Kip2 protein expression (Supplementary Fig. 6c). To further verify the effects of p57Kip2 on lactation, we detected the expression levels of *p57Kip2* during in vitro differentiation of HC11 cells. Results showed that *p57Kip2* expression was significantly increased at both the mRNA (Supplementary Fig. 6d) and protein levels (Fig. 5i), opposite to the FUS expression pattern observed during differentiation (Fig. 5h, i).

These findings suggest that FUS binds to the 3′UTR of *p57Kip2* mRNA, reducing mRNA stability and inhibiting p57Kip2 expression, thereby regulating the cell cycle of MEC. In addition to p57Kip2, FUS also regulated the expression of other cell cycle proteins, including p21, p27, Cyclin B1, Cyclin D1, and Cyclin E1 (Supplementary Fig. 6c), confirming the regulatory role of FUS in the cell cycle. RIP-qPCR assay confirmed that FUS could bind to the mRNA of cell cycle proteins (Supplementary Data 3 and Supplementary Fig. 6e), which may directly regulate their protein levels by regulating their mRNAs translation by interacting with sequence elements in the 5′ and 3′ untranslated regions[46,47].

### High FUS expression is correlated with human lactation deficiency

To further clarify the potential implications of FUS expression in human lactation, we collected fresh milk samples from 55 healthy lactating women within 5 days after giving birth to a full-term infant. mRNA expression in milk fat globules (MFGs) can serve as an indicator of gene expression in milk-secreting epithelial cells[1,48]. To ensure the exclusion of RNA contamination from immune cells, we assessed MFG mRNA levels of specific markers (Supplementary Fig. 6f). Among the diverse expression patterns of various RBPs, we observed a significant reduction in FUS levels in women with sufficient milk compared to those with insufficient milk (Fig. 6n). Thus, these findings indicate that *Fus* levels are negatively correlated with milk production in lactating women.

Mechanically, we discovered that the downregulation of FUS is necessary for MEC differentiation during the late pregnancy and lactation stages by stabilizing *p57Kip2* mRNA. Conversely, FUS overexpression during the lactation stage results in a reduction in *p57Kip2* mRNA stability and inhibition of MEC differentiation, leading to sparse alveolar distribution and negative regulation of lactation (Fig. 7).

## Discussion

Throughout mammalian evolution, there has been a progressive increase in the efficiency of milk production, serving as a vital carrier of intergenerational energy transfer, from marsupials to eutherians[3]. The composition of milk is complex, and its synthesis and secretion in eutherians begin during gestation in a tightly regulated process[49,50]. In the current study, we noted the existence of milk components in sugar gliders during the pubertal stage (Fig. 1e–h), suggesting that lactation regulation in sugar gliders may exhibit inefficiencies that emerged early in the evolution of mammary glands. This evolutionary shift prompted us to investigate the key factors driving the evolution of lactation.

Notably, we observed the absence of a GS-sequence in FUS from mammals harboring strong lactation ability (Fig. 2d and Supplementary Fig. 2). In contrast, this sequence was retained in marsupials known to exhibit lower milk production efficiency[3]. Based on serial findings (Fig. 2 and Supplementary Fig. 1), we concluded that the absence of the GS-sequence in FUS during mammary gland evolution resulted in low expression of FUS during lactation, thereby promoting high milk production efficiency. For the mechanism by which GS-sequence regulates the expression levels of FUS, various studies have emphasized the importance of strict regulation of mRNA stability in controlling gene expression[51,52]. Through mRNA stability experiments, we discovered that the insertion of GS-sequence enhanced the mRNA stability of *Fus*, suggesting that this may directly influence FUS expression levels (Fig. 2 and Supplementary Fig. 1). Furthermore, why does the insertion of GS-sequence affect mRNA stability? The stability of mRNA is primarily determined by its nucleotide sequence, which not only shapes its secondary and tertiary structure but also determines its accessibility by various RBPs[53,54]. In addition, RNA modification can affect the activity, localization, and stability of mRNA[55–57]. Mutations in the nucleotide sequence of *Fus* during mammary gland evolution may have influenced its RNA modification, thereby altering mRNA stability. However, further experiments are required to verify this hypothesis. In addition to the potential changes in RNA modification, GS-sequence may also impact FUS expression by affecting the multidimensional structure of FUS and regulating its stability and degradation[58,59].

When overexpressed FUS in MECs during the lactation stage, the mice exhibited a distinct phenotype characterized by sparse alveolar structures, insufficient milk production, and subsequent lactation failure, leading to poor newborn survival (Figs. 3, 4 and Supplementary Fig. 3, 4). Notably, the negative impact of *Fus* overexpression was less pronounced in the second lactation (Fig. 3e and Supplementary Fig. 3j) compared to the first (Fig. 3c, d and Supplementary Fig. 3h, i). A potential explanation for this observation is the typically higher volume of milk produced in the second pregnancy (including colostrum and mature milk) compared to the first, as documented in mice[60], human[61], rabbits[62], sows[63], and cows[64]. Successive pregnancies significantly re-organize both the rate and pattern of mammary gland alveolar development[65] and the overall size of the mammary gland in mice[60], potentially contributing to the increase in milk production between the first and second lactations. In addition, pregnancy may alter the receptivity of the gland to pregnancy-related hormones,

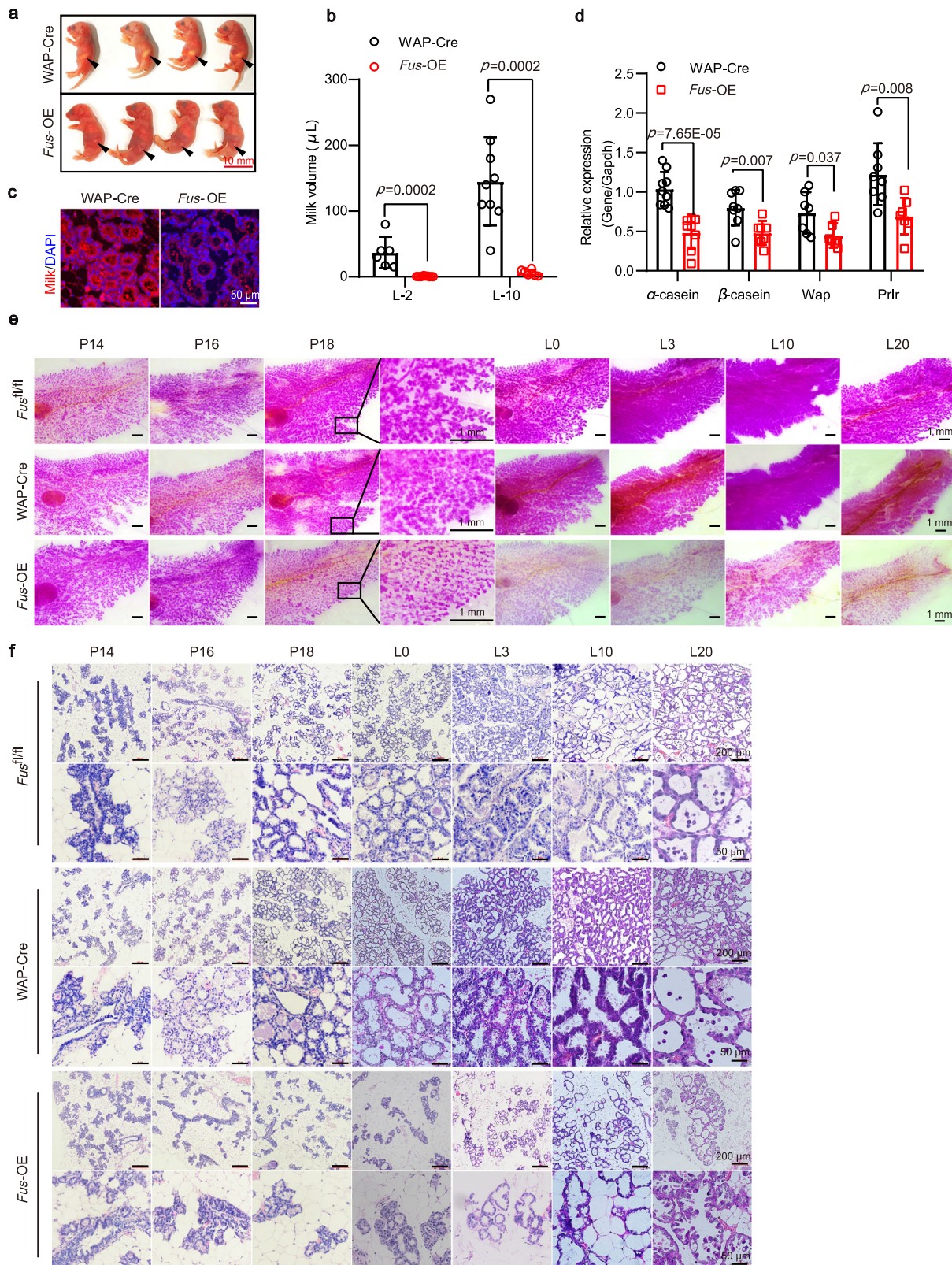

possibly through long-term epigenetic modifications. Genes affected by pregnancy-related epigenome changes are likely to be reactivated more rapidly in subsequent pregnancies[22].

The mammary epithelium is organized into a basal layer of myoepithelial cells and a luminal layer of secretory cells[11]. FUS expression was noted in both basal/myoepithelial cells and luminal cells (Supplementary Fig. 1f). Overexpression of FUS in luminal cells

(WAP-Cre system) resulted in impaired lactation, linked to changes in MEC proliferation and differentiation (Fig. 4e, f and Supplementary Fig. 3a). Similarly, overexpression of FUS in basal/myoepithelial cells (K14 system) led to a sparse alveolar phenotype (Supplementary Figs. 3a, 4f, g), but the detailed mechanism affecting lactation needs to be further explored. In addition to their role in contraction, basal/ myoepithelial cells have the potential to regulate alveolar cell

**Fig. 4 | High FUS expression induces insufficient milk production and sparse alveolar distribution. a** Gross appearance of neonates from WAP-Cre and *Fus*-OE females at 8 h postpartum. Representative stomach (arrowheads) of a neonate is shown. Scale bar: 10 mm. **b** Milk volume from mammary glands of *Fus*-OE and WAP-Cre mice following 10 U oxytocin injection at L2 (WAP-Cre: *n* = 6; *Fus*-OE: *n* = 10) and L10 (WAP-Cre: *n* = 9; *Fus*-OE: *n* = 6). **c** Immunofluorescence staining of milk (red) and 4′,6-diamidino-2-phenylindole (DAPI, blue) in mammary gland sections of WAP-Cre and *Fus*-OE mice at L10. Representative images from four mice are shown. Scale bar: 50 µm. **d** Relative mRNA expression of milk protein-related genes analyzed by RT-qPCR in mammary gland extracts from *Fus*-OE and WAP-Cre mice at L0.

WAP-Cre, *n* = 8 mice; *Fus*-OE, *n* = 7 mice. **e** Whole-mount carmine staining of mammary glands of *Fus*^fl/fl^, WAP-Cre, and *Fus*-OE mice at P14, P16, P18, L0, L3, L10, and L20. Magnified areas are shown in black boxes. The experiments were performed to have three biological replicates independently with similar results. Scale bar: 1 mm. **f** Representative H&E-stained sections of mammary glands from WAP-Cre and *Fus*-OE mice at P14 (*n* = 3 mice), P16 (*n* = 3 mice), P18 (*n* = 3 mice), L0 (*n* = 7 mice), L3 (*n* = 5 mice), L10 (*n* = 3 mice), and L20 (*n* = 3 mice). Scale bar: 200 µm (upper); 50 µm (bottom). Data were means ± SD. An unpaired *t*-test was used to evaluate statistical significance in graphs (**b**, **d**). *P* values are indicated in the chart (**b**, **d**). Source data are provided as a Source Data file.

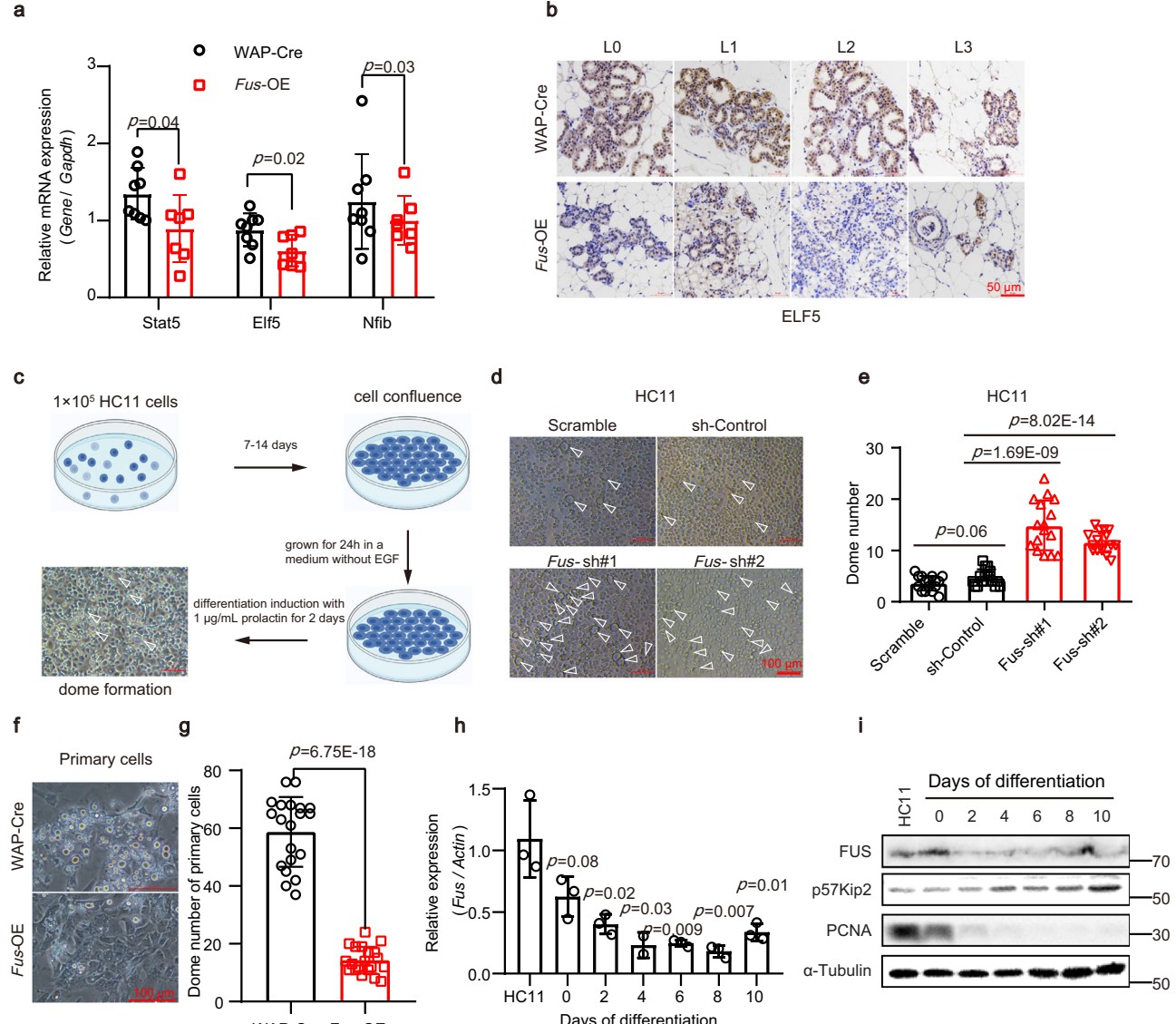

**Fig. 5 | FUS overexpression inhibits MEC differentiation. a** Relative mRNA expression of alveolar differentiation-related genes analyzed by RT-qPCR in mammary gland extracts from *Fus*-OE and WAP-Cre mice at L0. WAP-Cre, *n* = 8 mice; *Fus*-OE, *n* = 7 mice. **b** Immunohistochemical staining of ELF5 (brown) in mammary gland sections of WAP-Cre and *Fus*-OE mice at L0, L1, L2, and L3. Representative images from four mice are shown. **c** Abridged general view of dome formation during in vitro differentiation of HC11 cells treated with prolactin and other factors. The figure was created with BioRender.com. **d**, **e** Representative morphological images (**d**) and statistics (**e**) of dome formation during in vitro differentiation of HC11 cells expressed with indicated vectors. The arrowhead

indicates a dome structure. Three independent replicates were performed. 16 fields of view were used for statistics. **f**, **g** Representative morphological images (**f**) and statistics (**g**) of dome formation in mouse primary cell culture. *n* = 3 mice. 20 fields of view were used for statistics. **h** Relative mRNA expression of *Fus* during in vitro differentiation of HC11 cells at indicated time points. Scale bar: 100 µm. **i** Western blot analysis of protein expression levels at indicated time points after in vitro differentiation of HC11 cells. An unpaired *t*-test was used to evaluate statistical significance. Data were means ± SD of three independent experiments. The statistical test used was two-sided in graphs (**a**, **e**, **g**, **h**). *P* values are indicated in the chart (**a**, **e**, **g**, **h**). Source data are provided as a Source Data file.

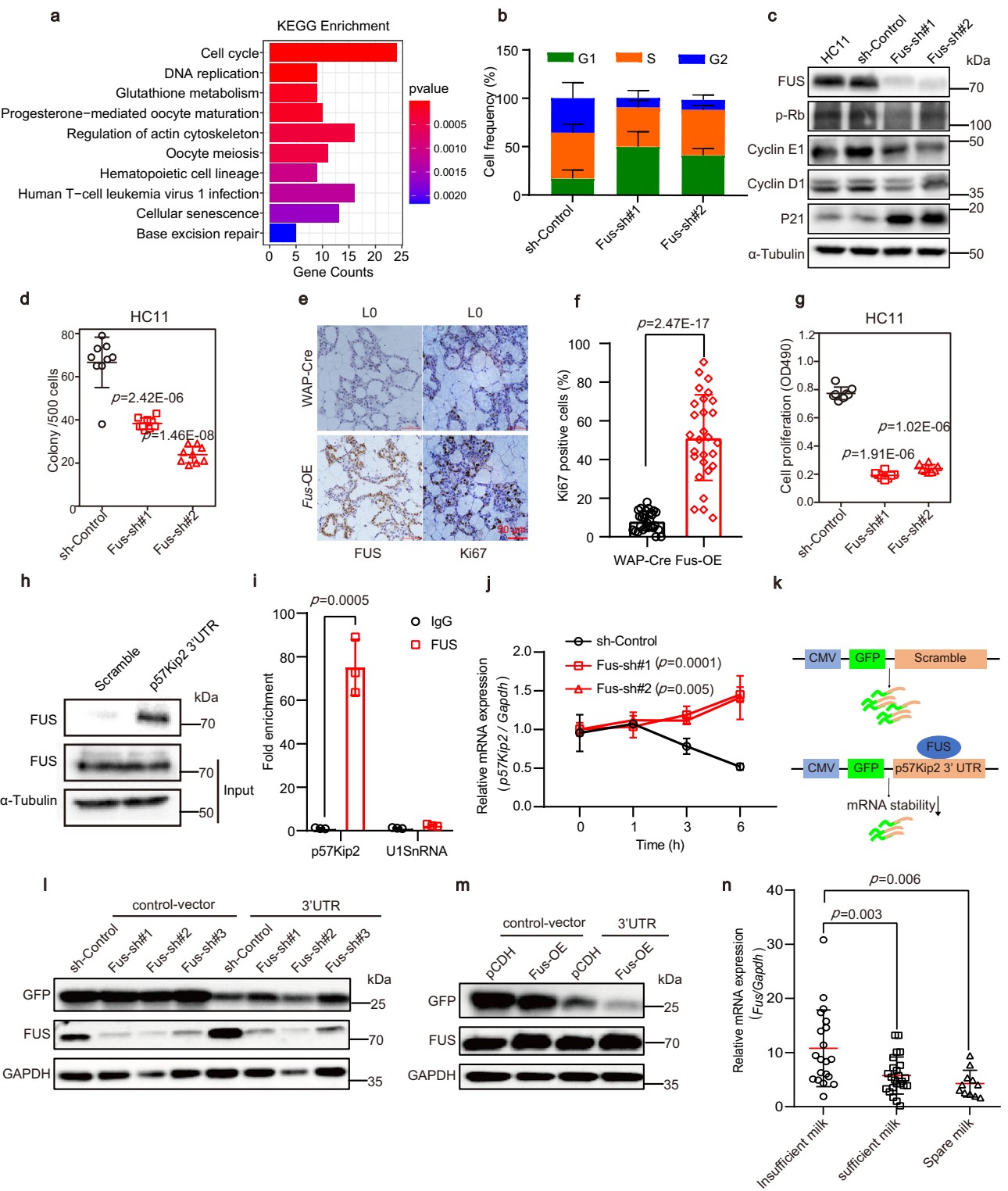

development via paracrine pathways[11]. For instance, the absence of p63 in basal/myoepithelial cells can lead to complete lactation failure due to obstructed cell proliferation and differentiation in the lumen[66]. The K5 promoter has been used to target the expression of stable N-terminal truncated β-catenin in the basal/myoepithelial cell layer of the mammary gland[67]. Activation of β-catenin signaling is accompanied by precocious lobular alveolar development during pregnancy, continuous proliferation, and accelerated involution of luminal cells during lactation[67]. In addition, the deletion of kindin-2[68] or the overexpression of JAG1[69] in basal/myoepithelial cells can regulate the

differentiation and lactation of alveolar cells by affecting Notch signaling pathway activation.

The mammary gland undergoes dynamic changes during different stages of development. In parallel, FUS also shows a dynamic pattern of expression. Our study revealed that the dynamic expression pattern of FUS plays a crucial role in coordinating the balance between MEC proliferation and differentiation (Figs. 5, 6 and Supplementary Figs. 5, 6). This coordination ensures appropriate cell cycle exit and differentiation of MEC by regulating the RNA stability of *p57Kip2* (Fig. 6 and Supplementary Fig. 6). Furthermore, to understand the regulatory

**Fig. 6 | FUS regulates coordination between cell cycle exit and MEC proliferation by directly binds to *p57Kip2* mRNA. a** KEGG analysis of transcriptome data from *Fus*-OE and WAP-Cre mice. $n = 3$. **b** Cell cycle phases (G1, S, and G2) in HC11 cells upon *Fus* knockdown. Data were means ± SD of three independent experiments. **c** Western blot analysis of indicated protein levels in HC11 cells upon *Fus* knockdown. **d** Colony number per 500 HC11 cells. The experiments were performed to have three biological replicates independently with similar results. **e**, **f** Immunohistochemical assay (**e**) and statistics (**f**) for FUS and Ki67 at L0. 24 fields of view from seven mice were used for statistics. **g** Analysis of HC11 cell proliferation upon *Fus* knockdown by MTS assay. The experiments were performed to have three biological replicates independently with similar results. **h** Analysis of FUS binding to *p57Kip2* mRNA 3'-UTR using RNA pull-down assay. **i** RIP-qPCR assay for analysis of the interaction between FUS protein and *p57Kip2* mRNA in HC11 cells. **j** RT-qPCR analysis of *p57Kip2* mRNA expression levels after treatment with actinomycin D for indicated times. **k** Schematic of reporter constructs containing GFP gene fused with *p57Kip2* mRNA 3'-UTR. **l**, **m** Western blot analysis of GFP expression levels after co-transfection with GFP reporter and *Fus*-sh (**l**) or *Fus*-OE (**m**) vector. **n** RT-qPCR analysis of *Fus* mRNA expression levels in fresh human milk on days 3–5 postpartum. Insufficient milk, $n = 20$; sufficient milk, $n = 24$; spare milk, $n = 11$. Data were means ± SD of three independent experiments. An unpaired *t*-test was used to evaluate statistical significance. The statistical test used was two-sided in graphs (**d**, **f**, **g**, **i**, **j**, **n**). P values were indicated in the chart (**d**, **f**, **g**, **i**, **j**, **n**). Source data are provided as a Source Data file.

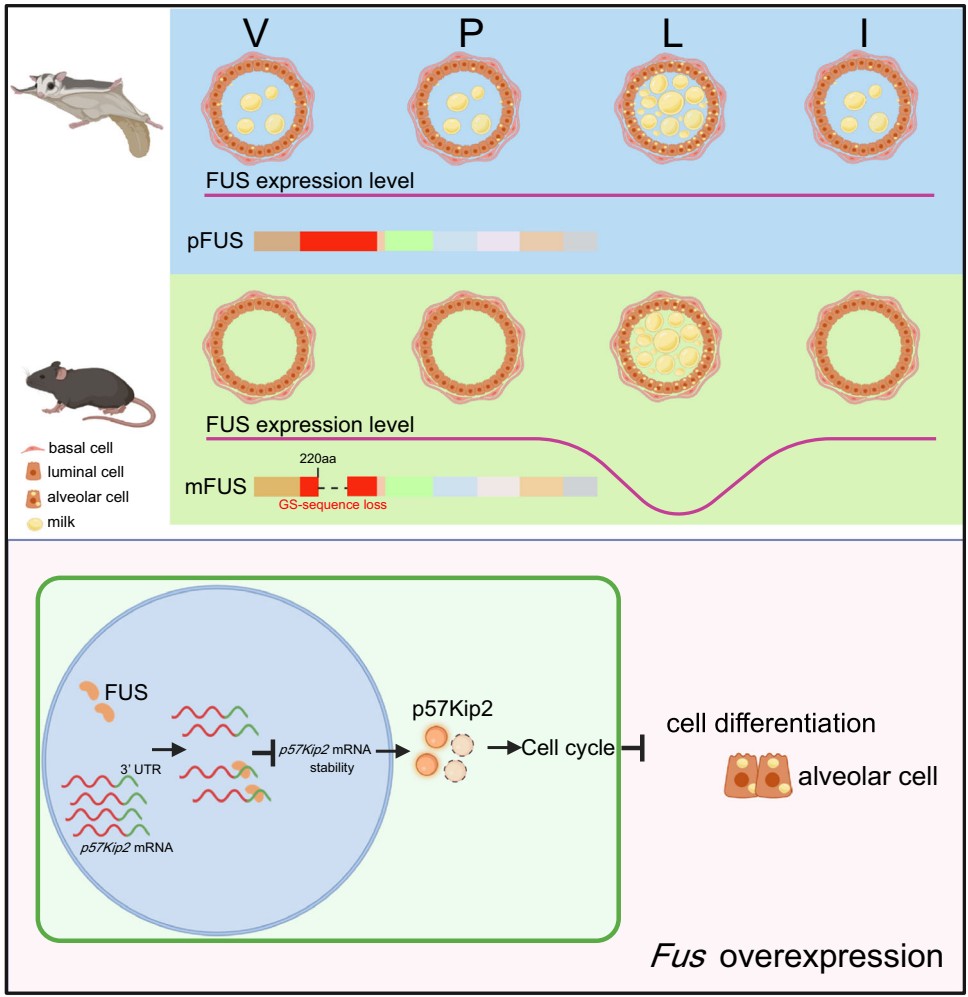

**Fig. 7 | Working model of roles of FUS in regulating milk production in mammals.** The figure was created with BioRender.com.

mechanisms underlying the dynamic expression pattern of FUS, we explored its hormonal regulation, considering the close relationship between mammary gland development and the endocrine system. Notably, previous studies in mice have shown that prolactin, progesterone, and estrogen regulate alveolar formation and secretion differentiation[27]. By re-analyzing previously published RNA-seq data[70], we found that exogenous progesterone injections had no significant effect on FUS expression in ovariectomized mice. Similarly, in vitro experiments treating cells with progesterone or oxytocin did not significantly impact FUS expression (Supplementary Fig. 7). These results suggest that low FUS expression during pregnancy is not hormonally regulated. Thus, further investigations are needed to unravel the underlying factors driving the dynamic expression pattern of FUS and its intricate relationship with mammary gland development.

In conclusion, we observed the gradual establishment of milk production from the marsupial to the eutherian development of the mammary gland. During this process, FUS was identified as a key negative regulator in mammary gland evolution. Clinically, the expression of *Fus* in human milk is inversely correlated with lactation ability (Fig. 6n). These findings enhance our understanding of mammary gland development and provide a new avenue for exploring the relationship between mammary gland evolution and the factors underlying inadequate breast milk production.

## Methods
### Animals
The wild-type (WT) mouse FUS coding sequence (CDS) (NM_139149.2) was inserted into the CAG-IRES-EGFP vector. The resulting construct

was purified and used for microinjection to generate *Fus* over-expression transgenic mice. The CAG-IRES-EGFP vector contains a CAG promoter followed by a loxP-flanked sequence and transcriptional STOP sequence, driving the expression of enhanced green fluorescent protein (EGFP)-coding regions linked by an internal ribosomal entry site (IRES) (Supplementary Fig. 3a). The inclusion of an IRES following the FUS open reading frame (ORF) enables independent translation of the EGFP ORF, thereby preventing the translation of the FUS-GFP fusion protein. To induce mammary gland-specific overexpression of FUS, the *Fus* overexpression transgenic mice were bred with WAP-Cre mice (stock #007905; Jackson Laboratory) and K14-Cre^ERT mice to yield *Fus*-OE mice. Tamoxifen (80 mg/kg/d) dissolved in ethanol and then in corn oil was administered to mice through intraperitoneal injection every other day for 3 days. All mice were bred and maintained under specific pathogen-free conditions. Genotyping was determined via PCR using the primers: *Fus*-LoxP: forward 5′-AGTCGCTCTGAGTTGT-TATCAG-3′, reverse 1 5′-TGAGCATGTCTTTAATCTACCTCGATG-3′, reverse 2 5′-AGTCCCTATTGGCGTTACTATGG-3′; Cre: forward 5′-AATGCTTCTGTCCGTTTGCCGG-3′, reverse 5′-CCAGGCTAAGTGCCT TCTCTACA-3′. The mice were raised in a specific pathogen-free (SPF) environment with an ambient temperature of 18–22 °C, a humidity of 50–60%, and a 12 h light-dark cycle. The sugar gliders were obtained from the pet trade and thereafter were maintained in breeding colonies at the Kunming Institute of Zoology, Chinese Academy of Sciences. Horseshoe bats were captured in Yunnan, China, between 2022 and 2023. Mice, sugar gliders, and horseshoe bats were euthanized by asphyxiation using a chamber with carbon dioxide and cervical dislocation as a secondary method. All experimental procedures and animal care and handling were performed per the protocols approved by the Ethics Committee of the Kunming Institute of Zoology, Chinese Academy of Sciences (IACUC-PA-2023-03-049).

### RNA sequencing and data analysis
Raw RNA sequencing data from mouse and sugar glider mammary glands in the virgin and lactation periods were processed through standard Illumina pipelines for base-calling and fastq file generation. Reads were then mapped against the genome assembly using HISAT2 v2.1.0[71]. The alignment results were converted to BAM format using SAMtools v1.9[72]. FeatureCounts v2.0.3[73] was used to assign sequence reads to annotation. Genes possessing less than 10 raw counts were removed. Differential expression analysis was performed with the Bioconductor DESeq2 package v1.34.0[74]. Significant genes in mice were determined by an adjusted $P$ value <0.05 and fold-change >2 or <−2. Non-significant genes in sugar gliders were determined by an adjusted $P$ value >0.05 and fold-change <−2 and <2. The reference genome and annotation versions used for analysis were GRCm39.108 for mice, Large_White_v1 for pigs, and ARS-UCD1.2 for cattle, with the sugar glider reference genome and annotation versions sourced from the FigShare repository: https://figshare.com/s/d6c585fbae0c1f22e8df[75].

### Genomic data
The FUS coding gene sequences (CDSs) of the ten species used in this study, including *Homo sapiens*, *Mus musculus*, *Capra hircus*, *Camelus bactrianus*, *Sus scrofa*, *Rhinolophus sinicus*, *Bos taurus*, *Petaurus breviceps*, *Macropus eugenii*, and *Sarcophilus harrisii*, were downloaded from the National Center for Biotechnology Information database (www.ncbi.nlm.nih.gov/nuccore).

### Whole-mount staining
The inguinal mammary glands were placed on a slide and fixed in 25% glacial acetic acid and 75% ethanol for 1–2 d. The tissue was then stained in carmine alum solution overnight at 4 °C, dehydrated with a graded series of ethanol solutions, cleared in xylene, and coverslipped with Neutral Balsam (Solarbio, #G8590).

### Histological analysis and immunostaining
Mammary glands were fixed in a 10% neutral-buffered formalin solution and embedded in paraffin. The paraffin-embedded tissue sections (5-μm thickness) were then dried at 65 °C for 1 h and stored at −20°C. The paraffin-embedded tissue sections were dehydrated with xylene and graded alcohol and boiled in 10 mM sodium citrate for 20 min for antigen retrieval. The sections were then used for immunofluorescence and immunohistochemical staining. For immunofluorescence, sections were blocked with 10% goat serum for 1 h, then incubated with primary antibodies overnight at 4 °C. The slides were then washed three times in phosphate-buffered saline (PBS) and incubated with secondary antibodies for 1 h at room temperature. The slides were counterstained with 4′,6-dia-midino-2-phenylindole dihydrochloride. For immunohistochemical analysis, sections were inactivated with endogenous peroxidases with 3% $H_2O_2$ for 10 min, then incubated with primary antibodies overnight at 4 °C and secondary antibodies for 1 h at room temperature after blocking for 1 h. Primary antibodies used for immunostaining were FUS (1:100, Abcam, #ab124923), FUS (1:800, Bethyl, #A300-294), CD133 (1:100, ABclonal, #A0219), EpCAM (1:100, Abcam, #ab71916), K14 (1:50, Abcam, #ab118685), K18 (1:50, Abcam, #ab133263), milk (1:500, Nordic Immunology, #5941), Ki67 (1:500, Abcam, #ab15580), ELF5 (1:500, ABclonal, #A7181), and E-cadherin (1:100, Abcam, #ab40772). Secondary antibodies used in immunostaining included Alexa Fluor 555 goat anti-rabbit IgG (1:500, Invitrogen, #A21428), Alexa Fluor 488 goat anti-mouse IgG (1:500, Abcam, #ab150113), and anti-rabbit IgG (1:200, Abcam, #ab6721).

### Raman spectroscopy analysis
Fresh breast tissues from mice and sugar gliders during virgin and lactation were frozen with the embedding agent (SAKURA, #4583) and then sliced. Milk secretion from mice and sugar gliders was evenly applied to the slides as a positive control for Raman spectroscopy. The slides are submitted to the company for Raman spectroscopic analysis. Then, the difference in Raman characteristic peaks between milk secretion and mammary gland contents inside the ductal cavity was compared.

### Oil Red staining
Fresh mammary tissue was placed in 4% paraformaldehyde and fixed at 4 °C. The fixed tissue was then dehydrated by soaking in a 30% sucrose solution for 24 h. After embedding the tissue in a tissue embedding medium (Surgipath, #3801480), 20-μm thick slices were prepared and stored at −80 °C. For staining, the slices were immersed in 60% isopropanol for 30 s, followed by incubation in prepared Oil Red working solution (Servicebio, #G1015) in the dark for 10 min. The slices were then taken out and immersed in 60% isopropanol for 5 s, repeating the process twice. After a brief double-distilled water rinse for 10 s, the slices were stained with hematoxylin. Finally, the slices were rinsed with running water and mounted with glycerol-alum (Servicebio, #G1402).

### Cell lines and culture
HC11 cells (obtained from the Bernd Groner Lab, Ludwig Institute for Cancer Research) were grown in RPMI-1640 medium (Gibco, #C11875500BT) with 10% fetal bovine serum (FBS), 5 μg/ml Insulin (Sigma, #I5500), 5 μg/ml gentamycin sulfate, and 10 ng/ml epidermal growth factor (EGF, Gibco, #PGH0315). The HEK293T cells were purchased and authenticated from the American Type Culture Collection (ATCC) and cultured in Dulbecco's Modified Eagle Medium (DMEM) (Gibco, #C11995500BT) supplemented with 10% FBS and 1% penicillin-streptomycin solution (PS). The MCF10A cells were purchased and authenticated from the ATCC and cultured in DMEM/F12 (Gibco, #C11330500BT) supplemented with 10% Horse Serum (HS), 20 ng/ml

EGF, 100 ng/ml Cholera toxin, 0.008 mg/ml Insulin, 50 ng/ml Hydrocortisone and 1% PS.

## RNA extraction and RT-qPCR

Total RNA was prepared using TRIzol reagent (Life Technologies), then converted to cDNA using a HiScript®III RT SuperMix for qPCR Kit (Vazyme, #R323-01). RT-qPCR analysis was performed using a SYBR qPCR Master Mix (Vazyme, #Q711-02) on a QuantStudio 3 instrument and normalized against GAPDH. Primers used are listed in Supplementary Table 2.

## Protein extraction and western blot analysis

Cell and breast tissue homogenates were subjected to lysis in RIPA buffer containing protease inhibitors. The samples were subjected to sodium dodecyl sulfate-polyacrylamide gel electrophoresis (SDS-PAGE) and transferred to a polyvinylidene difluoride (PVDF) membrane. The membranes were blocked with 5% nonfat dry milk for 1 h and incubated with primary antibodies at 4 °C overnight, then incubated with horseradish peroxidase (HRP)-linked secondary antibodies (Sigma) for 1 h at room temperature. A chemiluminescent HRP substrate (Millipore) was used to detect protein expression. The following primary antibodies were used for western blot analysis: α-Tubulin (1:5000, Sigma, #T5168), GAPDH (1:2000, Santa Cruz, #sc-25778), Flag (1:2000, CST, #14793), Cre (1:1000, Abcam, #ab190177), GFP (1:1000, Abcam, #ab290), FUS (1:1000, Abcam, #ab124923), p57Kip2 (1:1000, Abcam, #ab75974), PCNA (1:1000, Abcam, #ab29), p-RB (1: 50,000, ABclonal, #AP0484), Cyclin B1 (1:1000, Abcam, #ab72), Cyclin D1 (1:1000, CST, #2978 S), Cyclin E1 (1:1000, CST, #4129 S), P21 (1:1000, Abcam, #ab109199), and P27 (1:1000, BD biosciences, #610241). The secondary antibodies used were anti-mouse IgG (1: 5000, Abcam, #ab6728) and anti-mouse IgG (1:5000, Abcam, #ab6721).

## Plasmid construction, overexpression, and knockdown

For overexpression, full-length (FL) and mutant (GS-sequence inserted) human (NM_004960.4) or mouse (NM_139149.2) FUS CDS with Flag or GFP tags was cloned into pCDH-CMV-MCS-EF1-Puro lentiviral expression vector. For knockdown, shRNAs were cloned into the pLKO.1 vector. The shRNA sequences are listed in Supplementary Table 2. All plasmids were sequenced to rule out mutations. The vectors were co-transfected with psPAX2 and pMD2.G (4:3:1) into 293T cells to produce lentiviral particles, which were then transfected into HC11 or MCF10A cells. After 48–72 h, the cells were collected for further analysis or experiments.

## Primary MEC preparation

Breast tissue was cleaved and digested in lysate with DMEM/F12 containing 300 U/ml collagenase I (Sigma, #C0130), 100 U/ml hyaluronidase (Sigma, #H3506), 5% FBS, and 1% penicillin-streptomycin-glutamine for 1–3 h at 37 °C and 100 rpm. The cells were then treated with pancreatin for 5 min for dispersal, and DNA was removed with 5 mg/ml dispase (Sigma, #D4693) containing 0.1 mg/ml DNase I (Roche, #11248932001) for 5 min at 37 °C with gentle pipetting. Finally, red blood cells were removed in 0.8% NH4Cl, followed by filtration through a 40-mm filter. Primary MECs were grown in DMEM/F12 medium with 1 mM Glutamine, 5 μg/ml Insulin, 500 ng/ml Hydrocortisone, 10 ng/ml EGF, and 20 ng/ml Cholera toxin.

## mRNA stability analysis and differentiation induction

Cells were treated with 5 μg/ml actinomycin D (Sigma, #SBR00013) and collected at different times for RNA analysis to examine mRNA decay. For differentiation induction, a small number of cells were plated in six-well plates and allowed to reach confluence after 1–2 weeks. Once confluence was reached, cells were grown for 24 h in a medium without epidermal growth factor (EGF), followed by growth in

a medium containing 5 μg/ml prolactin (ProSpec, #cyt-240), 1 μM Dexamethasone, and 5 μg/ml Insulin.

## Mouse milk collection

Milk collection was performed according to previously reported methods[76]. Female mice were separated from their pups for 3–5 h at L2 or L10 and injected intraperitoneally with 0.2 U oxytocin (Sigma) after anesthesia. After 20 min, as much milk as possible was collected manually from each mammary gland nipple. Subsequently, milk-specific antibodies were utilized for staining to confirm that all milk had been completely collected from the mice. Samples in which milking was deemed incomplete, indicated by a substantial amount of milk remaining within the mammary gland, were excluded from further analysis.

## Cell cycle

HC11 and primary cells were cultured in a complete medium for 24 h at 37 °C after growth in a blank medium for 12 h at 37 °C. The cells were immobilized with 75% ethanol overnight at 4 °C, followed by incubation with propidium iodide (PI) mixture (0.1 mg/ml PI, 1 mg/ml RNase A, and 0.6% NP40) at 37 °C for 30 min away from light. The cells were then moved into a flow tube for cell cycle analysis by flow cytometry.

## Clone and MTS assays

Cells were grown in a medium containing Matrigel (Corning, #354234), and clone formation was observed and photographed after 10 days. For the MTS assay, primary and HC11 cells were incubated with an MTS mixture (Promega, #G5421) (MTS: blank medium = 1:5) at 37 °C for 1 h away from light after growth on a 96-well plate for 2 days. Absorbance at OD$_{490}$ was measured and analyzed.

## RNA immunoprecipitation

HC11 cells were collected and incubated with RIP lysis buffer overnight at −80 °C using the RIP Kit (Millipore, #17-701). The cell lysate was then immunoprecipitated with FUS (5 μg) (Bethyl, #A300-294A) and IgG (5 μg, Millipore, #CS200621) beads for 6–8 h at 4 °C, followed by washing with RIP buffer. RNA was then purified for RT-qPCR analysis and RIP-seq.

## RNA pull-down

The 3′UTR of p57Kip2 mRNA was synthesized in vitro using a HiScribe T7 High Yield RNA Synthesis Kit (NEB, #E2040S). Single biotinylated nucleotide was then attached to the RNA using a Pierce RNA 3′ End Desthiobiotinylation Kit (Thermo, #20163). The protein-RNA complexes were subsequently washed three times with wash buffer after the HC11 cell extract was incubated with biotin-labeled RNAs for 1 h at 4 °C using an RNA Pull-Down Kit (Thermo, #20164). Interaction between the FUS protein and 3′UTR of p57Kip2 mRNA were detected by protein elution and western blot analysis.

## Human milk sample collection and analysis

Written informed consent was obtained from all participants. Fresh milk (1–5 ml) was collected from 55 healthy females who gave birth to full-term babies 3–5 days after delivery and immediately placed on ice. The subsequent MFG separation steps were completed within 3 h before the sample was transferred to a −80 °C refrigerator. The upper-layer phase containing MFGs was carefully transferred to another RNase-free tube after centrifugation of the milk for 10 min at 13,000 rpm and 4 °C, followed by washing the MFGs with cold PBS three times. TRIzol lysis reagent was added to the MFGs for RNA extraction. Mothers were followed up by phone at 7–8 weeks postpartum to enquire about breastfeeding and formula feeding. According to the follow-up results, the samples were divided into three groups: insufficient milk (predominantly formula feeding), sufficient milk (meeting the baby's needs), and spare milk (more than the baby's needs). This

project was approved by the Ethics Committee of Weifang People's Hospital or Luoyang Maternal and Child Health Hospital.

## Statistics and reproducibility

All animals were randomly assigned to the experimental groups, except when the purpose of the experiment was to compare the difference between controls and *Fus*-OE mice. In histological staining, more than four visual fields in each of the five directions of the upper, lower, left, right, and middle were randomly selected for statistics. No data were excluded from the experiments. The investigators were not blinded to outcome assessments. All experiments were performed to have at least three biological replicates to ensure power for statistical analysis using a two-sided student *t*-test. Sample groups for all experiments were not blinded. *P* values equal to or <0.05 were considered to be significant. All immunoblots are performed at least three biological replicates unless specified in figure legends. Graphs and error bars reflect means ± SD. All statistical analyses were carried out using GraphPad Prism 8.0.

## Reporting summary

Further information on research design is available in the Nature Portfolio Reporting Summary linked to this article.

## Data availability

All data used in this study are available within the Article and Supplementary information, or available from the corresponding authors on request. Source data are provided as Source Data file. Source data are provided with this paper. The raw transcriptomics data produced and analyzed in this study have been deposited in the Genome Sequence Archive in the National Genomics Data Center, China National Center for Bioinformation / Beijing Institute of Genomics, Chinese Academy of Sciences that are publicly accessible at https://bigd.big.ac.cn/gsa/browse/CRA014528. The accession code is GSA: CRA014528. Source data are provided with this paper.

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

## Acknowledgements

This work was supported by the National Key Research and Development Program of China (2023YFA1800500, B.J.), National Natural Science Foundation of China (U23A20452, B.J., 31701306, H.W., and 82272727, H.W.), Yunnan Applied Basic Research Key Projects (202201AS070039, B.J., and 202401AS070079, H.W.), Yunnan Applied Basic Research Projects (202301AT070280, H.W.), Spring City Project from Kunming Science and Technology Bureau (2022SCP007, B.J.), CAS "Light of West China" Program (H.W.) and Yunnan Revitalization Talent Support Program Young Talent Project (H.W.). We would like to thank Nengyin Sheng and Ceshi Chen for constructive suggestions and Christine Watts for English editing. We thank the Institutional Center for Shared Technologies and Facilities of Kunming Institute of Zoology (KIZ), Chinese Academy of Sciences (CAS) for providing us with Confocal Microscopy image acquisition and flow cytometric analysis from Cong Li, Guolan Ma, and Xiuzhen Yang.

## Author contributions

B.J., H.S., J.H., and H.W. conceived the project, designed the experiments, and wrote the manuscript. H.S., H.W., J.H., and M.C. conducted all experiments. H.S., H.W., J.H., M.C., X.Y., P. S., B.H., and L.S. discussed or analyzed the data. H.W., G.W., D.Z., and B.H. collected human milk

and conducted postpartum follow-up. C.S., Z.L., and X.J. helped obtain horseshoe bat samples. L.Z. and Q.Y. purchased the reagents.

## Competing interests

The authors declare no competing interests.
