## [Peer Review File · Nature Communications]

Fused in sarcoma (FUS) inhibits milk production efficiency in mammalsREVIEWER COMMENTS

Reviewer #1 (Remarks to the Author):

Review on the manuscript "Fused in sarcoma (FUS) inhibits milk production efficiency in mammals" by Shao and co-workers. The manuscript describes an unexpected finding for those who work on FUS, that FUS might regulate milk production by regulating cell proliferation and differentiation of mammary gland cells. While the topic is novel, the molecular analysis needs additional work to substantiate the findings which would justify many of the statements in the manuscript.

In detail:

M&M section is much too short, one cannot properly follow the single experiments, and thus cannot judge whether these have been made properly (which I assume). This starts with more details on the mouse generation, the check for putative off target effects in the Cre recombination. There is also not analysis of the FUS lox mice, whether those also already show a phenotype (due to random cre expression etc.) and goes through all experiments.

Fig. 2H: Actinomycin D treatment : be more specific, it blocks RNA synthesis, and often used to then measure the protein level to measure protein turnover/longevity of the protein thus the mRNA results shown suggest that amount of FUS mRNA is longer present, which suggests higher stability, which might be due to lack of degradation but also due to a lack of transcription.

◇would also need to measure protein expression over time.

Fig 3: Mice models: more details on the mouse generation, the check for putative off target effects in the Cre recombination (leakiness in the brain/spinal cord). There is also not analysis of the FUS lox mice, whether those also already show a phenotype (due to random cre expression etc.)

FUS is tightly regulated in organisms, overexpression of wildtype FUS can cause motor neuron disease like phenotypes. Thus this must be ruled out as a possible cause of death as well.

Furthermore, would need more details on the CAG-IRES-EGFP construct, were exactly is the FUS, it is N- or C-terminally tagged by the gfp? There are several reports showing that gfp tagging of FUS can induce cytoplasmic mislocalization and thus mimicking amyotrophic lateral sclerosis phenotypes. Thus a FUS staining with high magnification is needed to show whether there is artificial cytoplasmic GFP localization.

Fig 3G: Not decrease in weight but decreased weight gain

Fig 3F: any significances?

“Although the newborns appeared to suckle normally, their stomachs contained minimal milk (Fig. 4A).”

◇how to distinguish from muscle weakness (ALS like phenotype) with impaired suckling?

General aspect on figures: Figure are somehow difficult to follow because of unusual ordering (e.g. Fig 5)

Fig 5: Don't understand why for the dome like structure experiments in HC11 only FUS knockdown was performed, the FUS overexpression in primary mammaria cells, but not both in both cell lines?

Fig 6B: cannot see an arrest, rather a G1/G0 enlargement, which in FACS cannot be distinguished between asymmetric division and more cells entering G0 which means differentiated cells.

Clone forming assay and pluripotency: . . .”impaired clone formation ability (Fig. 6C and Extended Data Fig. 5B, C), suggesting that FUS contributes to pluripotency maintenance.”

◇without showing differentiation capacity after clone formation, one can only talk about colony forming, but not pluripotency!

“Taken together, these findings indicate that high FUS expression inhibits MEC differentiation by inhibiting cell cycle exit.”

◇the author showed that FUS-OE increased, FUS knockdown decreased proliferation. They also showed enlarged G1 population on the cost of G2. This data together only show the existence of more differentiating cells. The authors did not properly investigate cell cycle

exist by e.g. analyzing checkpoints etc., which would be necessary to state something like this.

Instead, as the authors say themselves “Alveolar-like cells can form milk-like dome-shaped structures, which can serve as indicators of MEC differentiation.”, they showed in fig. 5, that FUS-OE led to decreased, FUS knockdown to increased differentiation.c.

Thus it seems for me that the cells switch from proliferation to differentiation, so most likely enter G0 properly. This has to be analyzed in much more detail, or this statement on the cell cycle cannot be made!

“These findings suggest that FUS binds to the 3’UTR of p57Kip2 mRNA, reducing mRNA stability and inhibiting p57Kip2 expression, thereby regulating the cell cycle of MEC.”

◇the data and experiments on p57Kip2 are convincing, but once more: a much more in depth analysis of the protein landscape of the cell cycle proteins/checkpoints are needed, and putative role of FUS on those as well, to really judge on the changes in the cell cycle regulation.

Reviewer #2 (Remarks to the Author):

This manuscript has been shown that fused in sarcoma (FUS) inhibits milk production efficiency. I think this research is very interesting and this conclusion is appropriate. However, I have other thoughts on the interpretation of the results that lead me to this conclusion. My interpretation about the data in Fig. 4-6 is that FUS inhibits mammary alveolar formation by affecting cell cycle in late pregnancy, resulting in fewer mammary follicles and low milk production during lactation. There are no histological data in late pregnancy in the current manuscript. I would strongly recommend examining how mammary alveolar formation is suppressed in late pregnancy (at least HE staining images about them). There are also some things that needed to be added or modified before acceptance.

L44-L49

The lactation periods of mice, dog, humans, and cows are different even if the same eutherian species. Basically, the lactation period length is shorter for smaller animals and

longer for larger ones. Therefore, when comparing lactation period lengths of marsupials and eutherians, species of animals of similar weight should be selected, respectively, in the manuscript.

L66

Fill in the full name first, not the abbreviation. This is first of "FUS" in the manuscript.

L59-75

"Introduction" should refrain from describing excessive results.

L84 (Fig.1)

To make valid comparison, the detailed information such as age, sexual cycle (If sugar gliders had one), days of pregnancy, and involution day after weaning should be added in in Method or Figure caption.

L84 (Fig. 1B)

Fig.1B is a mammary duct but not a mammary alveolus.

L87-89 (Fig. 1C)

It is difficult to distinguish between the mammary alveoli and ducts during lactation by HE staining. Especially the peripheral mammary ducts are about the same size as the mammary alveoli. To improve the paper, it is recommended to identify mammary ducts by immunostaining of breast duct marker molecules.

L102-108 (Fig.1 1H)

It would be good to show what milk component each peak indicates in graphs.

L266-272 (Fig.5C-G)

There is no causal relationship between dome formation and milk production. Dome formation occurs by changing in cell adhesion signaling. In this case, the expression levels of milk proteins in HC11 cells should be examined.

L208 (Fig. 3G)

Please explain in Discussion why the adverse effects of Fus-overexpression decrease in the second lactation than the first lactation.

L233-246 (Fig. FE-H)

CK14 and WAP are used for Fus overexpression in the experiments. However, WAP is expressed in alveolar luminal epithelial cells whereas CK14 is expressed in basal cells/myoepithelial cells. Therefore, the cell types that are Fus-overexpressed are different. However, the results were the almost same in the mice using CK14 and WAP systems. Please explain about them in Discussion.

Reviewer #1 (Remarks to the Author):

Review on the manuscript “Fused in sarcoma (FUS) inhibits milk production efficiency in mammals” by Shao and co-workers. The manuscript describes an unexpected finding for those who work on FUS, that FUS might regulates milk production by regulating cell proliferation and differentiation of mammary gland cells. While the topic is novel, the molecular analysis needs additional work to substantiate the findings which would justify many of the statements in the manuscript.

Response: Thank you for your positive comments and suggestions. We have modified the manuscript as per your suggestions, as detailed below.

In detail:

M&M section is much too short, one cannot properly follow the single experiments, and thus cannot judge whether these have been made properly (which I assume). This starts with more details on the mouse generation, the check for putative off target effects in the Cre recombination. There is also not analysis of the FUS lox mice, whether those also already show a phenotype (due to random cre expression etc.) and goes through all experiments.

Response: Thank you for your suggestions to improve the manuscript. The Materials and Methods section has been described in more detail. For FUS transgenic mice, the CAG-IRES-EGFP vector contains a CAG promoter, followed by a loxP-flanked sequence and transcriptional STOP sequence, which drives the expression of enhanced green fluorescent protein (EGFP)-coding regions linked by an internal ribosomal entry site (IRES) (Fig. S3A). The inclusion of an IRES following the FUS open reading frame (ORF) enables independent translation of the EGFP ORF, thereby preventing the translation of the FUS-EGFP fusion protein.

S3A

S3C

4E

4F

Figures S3A, S3C, 4E, and 4F. (S3A) Mating strategy of *Fus*-OE mice. Mice homozygous for a floxed allele of *Fus* ($Fus^{fl/fl}$ mice) were crossed with WAP-Cre or K14-Cre^{ERT2} transgenic mice. Pregnant ($Fus^{fl/fl}$ /K14-Cre^{ERT2}) mice were injected intraperitoneally with 240 mg/kg of tamoxifen/mouse body weight, divided into three doses once every other day. Tamoxifen was dissolved in

sunflower oil containing 10% ethanol at a final concentration of 20 mg/ml according to previous reports ^{1, 2}. (S3C) Western blot analysis of Cre, FUS, and GFP expression levels in mammary glands in WAP-Cre (n = 4 mice), *Fus*-OE (n = 5 mice), and *Fus^{fl/fl}* (n = 4 mice) mice. (4E) Whole-mount carmine staining of mammary glands of *Fus^{fl/fl}*, WAP-Cre, and *Fus*-OE mice at indicated stages. Magnified areas are shown in black boxes. Scale bar: 1 mm. (4F) Representative H&E-stained sections of mammary glands from *Fus^{fl/fl}*, WAP-Cre, and *Fus*-OE mice at P14 (n = 3 mice), P16 (n = 3 mice), P18 (n = 3 mice), L0 (n = 7 mice), L3 (n = 5 mice), L10 (n = 3 mice), and L20 (n = 3 mice). Scale bar: 100 μ m (upper); 50 μ m (bottom).

Thank you again for your critical consideration of the off-target effects in FUS mice. To address this concern, we first examined the expression levels of FUS and their effects on mammary gland development with an additional control group (FUS lox mice or *Fus^{fl/fl}* mice). Compared to the *Fus*-overexpressed mice (*Fus^{fl/wt}*/WAP-Cre or *Fus*-OE) and WAP-Cre mice, Cre recombinase, FUS, and GFP were undetectable in the *Fus^{fl/fl}* mice (Fig. S3C), confirming the absence of unintended expression of these proteins in the *Fus^{fl/fl}* mice. In addition, while whole-mount analysis revealed notable atrophy and sparsity of secretory lobuloalveolar structures in *Fus*-OE mammary glands, the *Fus^{fl/fl}* group showed similar structures to the WAP-Cre group (Fig. 4E). This observation was further confirmed by H&E staining, which validated the comparable density of alveolar structures in the *Fus^{fl/fl}* group (Fig. 4F). These findings suggest that the FUS lox mice do not exhibit off-target effects.

We also performed additional experiments to exclude possible off-target effects of Cre recombinase, as detailed on Pages 5-6 below (Fig. S3E-S3F).

Fig. 2H: Actinomycin D treatment : be more specific, it blocks RNA synthesis, and often used to then measure the protein level to measure protein turnover/longevity of the protein thus the mRNA results shown suggest that amount of FUS mRNA is longer present, which suggests higher stability, which might be due to lack of degradation but

also due to a lack of transcription.

◇ would also need to measure protein expression over time.

Response: Thank you for your suggestions. We examined the FUS protein levels after actinomycin D treatment upon the insertion of wild-type (hFUS) and GS sequences (hFUS-PGS, hFUS-MGS, and hFUS-SGS) at various timepoints (0, 6, 12, 18, 24, and 36 h) according to the previous reports^{3,4} by western blotting analysis. Results showed that the FUS levels in the wild-type (hFUS) began to decrease at 24 h. However, in the GS-insertion variants, the protein levels persisted for a longer duration under actinomycin treatment (24-36 h) (Fig. S1K). This suggests that the GS sequence significantly inhibited the down-regulation of FUS protein levels. This, combined with our findings at the mRNA level (Fig. 2F-2H, and S1H-S1J), implies that the GS sequence contributes to enhanced stability of FUS mRNA, thereby slowing its degradation and sustaining higher levels of FUS protein expression.

S1K

Figure S1K. Western blot analysis of Flag-FUS protein levels in HC11 cells treated with actinomycin D after overexpression of pCDH-3xFlag (pCDH), FUS full-length (hFUS), and FUS mutants (hFUS-PGS, hFUS-MGS, and hFUS-SGS).

Fig 3: Mice models: more details on the mouse generation, the check for putative off target effects in the Cre recombination (leakiness in the brain/spinal cord). There is also not analysis of the FUS lox mice, whether those also already show a phenotype (due to random cre expression etc.)

FUS is tightly regulated in organisms, overexpression of wildtype FUS can cause motor neuron disease like phenotypes. Thus this must be ruled out as a possible cause of death as well.

Response: Thank you for your suggestion. We isolated and examined various brain areas (including the cerebral cortex, hippocampus, and limbic system (excluding the hippocampus) and spinal cord to examine the expression of FUS and Cre recombinase by western blotting. Results indicated that all isolated samples were negative for Cre expression (mammary gland of *Fus*-OE mice, MG, as a positive control), suggesting that Cre did not leak into the brain and spinal cord.

We also observed GFP expression exclusive to the *Fus*-OE mice (Fig. S3C), as verified in the positive control (MG). All other samples showed no GFP expression. Additionally, FUS levels were normal compared to the control sample, indicating that there was no overexpression of FUS (Fig. S3E, S3F).

These findings effectively exclude any unintended activity (leakiness) of Cre recombinase in the brain and spinal cord, as well as any off-target effects in the FUS floxed mice.

S3E

S3F

Figures S3E and S3F. (S3E-S3F) Western blot analysis of Cre, FUS, and GFP expression levels in brain (including cerebral cortex and limbic system) (S3E) and spinal cord (S3F) in WAP-Cre, *Fus*-OE, and *Fus^{fl/fl}* mice (n = 4 or 6 mice).

MG: mammary gland tissue of *Fus*-OE mice.

Furthermore, would need more details on the CAG-IRES-EGFP construct, were exactly is the FUS, it is N- or C-terminally tagged by the gfp? There are several reports showing that gfp tagging of FUS can induce cytoplasmic mislocalization and thus mimicking amyotrophic lateral sclerosis phenotypes. Thus a FUS staining with high magnification is needed to show whether there is artificial cytoplasmic GFP localization.

Response: Thank you for your questions and suggestions. Regarding FUS transgenic mice, the placement of IRES after the FUS open reading frame (ORF) (Fig. S3A) ensures the independent translation of the EGFP ORF, thereby preventing the translation of the FUS-EGFP fusion protein. According to your suggestion, we performed high-magnification immunofluorescence (IF) staining to determine the location of the FUS. Results showed that FUS was predominantly expressed in the nucleus (Fig. S3D), excluding the potential for cytoplasmic mislocalization of FUS in the *Fus*-OE mice.

Figures S3A and S3D. (S3A) Mating strategy of *Fus*-OE mice. Mice homozygous for a floxed allele of *Fus* (*Fus*^{fl/fl} mice) were crossed with WAP-Cre or K14-Cre^{ERT} transgenic mice. Pregnant (*Fus*^{fl/fl}/K14-Cre^{ERT}) mice were

injected intraperitoneally with 240 mg/kg tamoxifen/mouse body weight, divided into three doses once every other day. Tamoxifen was dissolved in sunflower oil containing 10% ethanol at a final concentration of 20 mg/ml^{1, 2}. (S3D) Immunofluorescence staining of mammary glands of *Fus*-OE mice for FUS (red) and DAPI (blue) at lactation. Scale bar: 50 μ m.

Fig 3G: Not decrease in weight but decreased weight gain

Response: Thank you for your question. We compared the weight gain of pups in the *Fus*-OE and WAP-Cre groups and found that weight gain was significantly lower in the *Fus*-OE group than in the WAP-Cre group (Fig. S3K).

Figure S3K. Weight gain of pups after litter size adjustment to seven on L2 during second lactation. Data are means \pm SD. Unpaired *t*-test was used to evaluate statistical significance. *, $P < 0.05$; ****, $P < 0.0001$.

Fig 3F: any significances?

Response: Thank you for your suggestion. We have marked significance levels in the revised manuscript (Fig. 3F).

Figure 3F. Body weights of surviving pups nursed by *Fus*-OE vs. WAP-Cre mothers. Body weights of surviving pups nursed by biological mother (*Fus*-OE or WAP-Cre, black lines) and cross-nursed by foster mother (WAP-Cre or *Fus*-OE) at L2 (red lines) and L10 (blue lines), recorded at the beginning of L2. Litter size was adjusted to six for each foster mother on L2. Data are means \pm SD. Unpaired *t*-test was used to evaluate statistical significance. *, $P < 0.05$; **, $P < 0.01$; ***, $P < 0.001$; ****, $P < 0.0001$; ns, no significant difference.

“Although the newborns appeared to suckle normally, their stomachs contained minimal milk (Fig. 4A).”

◇ how to distinguish from muscle weakness (ALS like phenotype) with impaired suckling?

Response: Thank you for your suggestion. To exclude the possibility of muscle weakness affecting suckling ability, we conducted cross-fostering experiments. For the pups given birth by *Fus*-OE mothers, 1/3 of the pups were fed by the foster mothers of WAP-Cre genotype or *Fus*^{fl/fl} genotype on day 1, respectively. The breastfeeding status was observed after 8 h of cross-fostering. Results demonstrated that the stomachs of pups born to *Fus*-OE mothers but fostered by WAP-Cre mice or *Fus*^{fl/fl} mice contained

sufficient milk, implying that pups born to *Fus*-OE mothers had normal suckling abilities (Fig. S4A). Conversely, as a control, pups born to WAP-Cre mothers but fostered by *Fus*-OE mice contained minimal milk in their stomachs (Fig. S4B). These results suggest that the pups of both *Fus*-OE and WAP-Cre genotypes could suckle milk normally, thereby excluding impaired suckling as a contributing factor to their mortality. To describe the data accurately, we deleted the sentence “Although the newborns appeared to suckle normally” in the manuscript.

Figures S4A and S4B. (S4A-S4B) Gross appearance of neonates born to *Fus*-OE (S4A) and WAP-Cre mothers (S4B) but fed by WAP-Cre, *Fus*^{fl/fl}, or *Fus*-OE mothers. Representative stomach (arrowheads) of neonate is shown. Scale bar: 10 mm.

General aspect on figures: Figure are somehow difficult to follow because of unusual ordering (e.g. Fig 5)

Response: Thank you for your suggestion. We adjusted the order of the figures.

Fig 5: Don't understand why for the dome like structure experiments in HC11 only FUS knockdown was performed, the FUS overexpression in primary mammary cells, but not both in both cell lines?

Response: Thank you for your suggestion. Accordingly, we carried out *Fus*

overexpression in HC11 cells and found that dome formation was significantly inhibited (Fig. S5C, S5D). In addition, dome formation was significantly increased upon *Fus* knockdown in primary cells (Fig. S5E, S5F). These results strongly demonstrate that FUS inhibits the differentiation of mammary epithelial cells.

Figures S5C-S5F. (S5C-S5D) Representative morphological images (S5C) and statistics (S5D) of dome formation during *in vitro* differentiation of HC11 cells expressed with indicated vectors. Arrowhead indicates dome structure. (S5E-S5F) Representative morphological images (S5E) and statistics (S5F) of dome formation in mouse primary cell culture upon *Fus* knockdown. Data are means \pm SD. Unpaired *t*-test was used to evaluate statistical significance. ****, $P < 0.0001$.

Fig 6B: cannot see an arrest, rather a G1/G0 enlargement, which in FACS cannot be distinguished between asymmetric division and more cells entering G0 which means differentiated cells.

Response: Thank you for your careful reading and valuable comments. We have added new supporting evidence for our findings on cell cycle arrest. In addition to the observed increase in cell proportion in the G0/G1 phase upon *Fus* knockdown (Fig. 6B), we examined the levels of positive regulators that facilitate G1/S transition, including retinoblastoma tumor suppressor (Rb), Cyclin E, and Cyclin D^{5,6,7}. The levels of these

proteins were markedly decreased, while negative regulators, e.g. P21⁸, were markedly increased (Fig. 6C). These data suggest that the cell cycle is arrested at the G0/G1 phase upon *Fus* depletion.

Figure 6C. Western blot analysis of indicated protein levels in HC11 cells upon *Fus* knockdown.

Clone forming assay and pluripotency: . . .”impaired clone formation ability (Fig. 6C and Extended Data Fig. 5B, C), suggesting that FUS contributes to pluripotency maintenance.”

◇ without showing differentiation capacity after clone formation, one can only talk about colony forming, but not pluripotency!

Response: Thank you for your suggestion. We deleted the sentence “suggesting that FUS contributes to pluripotency maintenance”.

“Taken together, these findings indicate that high FUS expression inhibits MEC differentiation by inhibiting cell cycle exit.”

◇ the author showed that FUS-OE increased, FUS knockdown decreased proliferation. They also showed enlarged G1 population on the cost of G2. This data together only show the existence of more differentiating cells. The authors did not properly investigate cell cycle exit by e.g. analyzing checkpoints etc., which would be necessary to state something like this.

Instead, as the authors say themselves “Alveolar-like cells can form milk-like dome-shaped structures, which can serve as indicators of MEC differentiation.”, they showed in fig. 5, that FUS-OE led to decreased, FUS knockdown to increased differentiation.c. Thus it seems for me that the cells switch from proliferation to differentiation, so most likely enter G0 properly. This has to be analyzed in much more detail, or this statement on the cell cycle cannot be made!

Response: Thank you very much for your valuable comments. We agree that while the evidence supporting cell differentiation is robust, additional data are indeed necessary to substantiate the claim regarding the existence of cell cycle involvement.

As suggested, our new data showed that the expression of negative cell cycle regulators (P21)⁸ was up-regulated, while the expression of positive cell cycle regulators (p-RB, Cyclin D1, and Cyclin E1)⁹ was down-regulated upon *Fus* knockout, resulting in cell cycle arrest at the G1 phase (Fig. 6C). Elongation of the G1 phase can potentially facilitate the accumulation of transcription factors that trigger differentiation, as indicated in previous studies¹⁰. In line with this, our earlier data showed a decrease in the expression of differentiation-related transcription factors in *Fus*-OE mice (Fig. 5A). These findings suggest that a reduction in FUS levels leads to an extended G1 phase, thereby accumulating differentiation-related transcription factors. This, in turn, supports the transition of cells from a state of proliferation to differentiation, aligning with findings from prior research^{11,12}. Consequently, we propose that FUS may impede MEC differentiation by restricting cell cycle exit.

Figures 6C and 5A. (6C) Western blot analysis of indicated protein levels in

HC11 cells upon *Fus* knockdown. (5A) Relative mRNA expression of alveolar differentiation-related genes analyzed by RT-qPCR in mammary gland extracts from *Fus*-OE and WAP-Cre mice at L0. WAP-Cre, n = 8 mice; *Fus*-OE, n = 7 mice. Data are means \pm SD of three independent experiments. Unpaired *t*-test was used to evaluate statistical significance. *, $P < 0.05$; ***, $P < 0.001$.

“These findings suggest that FUS binds to the 3’UTR of p57Kip2 mRNA, reducing mRNA stability and inhibiting p57Kip2 expression, thereby regulating the cell cycle of MEC.”

∅ the data and experiments on p57Kip2 are convincing, but once more: a much more in depth analysis of the protein landscape of the cell cycle proteins/checkpoints are needed, and putative role of FUS on those as well, to really judge on the changes in the cell cycle regulation.

Response: Thank you for your suggestion. We investigated the protein levels of various cell-cycle proteins. In addition to p57Kip2, FUS also regulated the expression of other cell-cycle proteins, including p21, p27, Cyclin B1, Cyclin D1, and Cyclin E1 (Fig. S6C), further confirming the cell cycle regulatory role of FUS. In addition, RIP-qPCR assay confirmed that FUS can bind to the mRNA of cell cycle proteins (Supplementary Data 3, and Fig. S6E), which may directly regulate their protein levels by regulating their mRNAs translation by interacting with sequence elements in the 5’ and 3’ untranslated regions^{13, 14}.

Figures S6C and S6E. (S6C) Western blot analysis of indicated protein levels in HC11 cells upon *Fus* knockdown. (S6E) RIP-qPCR analysis of interaction between FUS protein and indicated mRNA in HC11 cells. U1SnRNA was applied as a negative control. Data are means \pm SD of three independent experiments. Unpaired *t*-test was used to evaluate statistical significance. ****, $P < 0.001$.

Reviewer #2 (Remarks to the Author):

This manuscript has been shown that fused in sarcoma (FUS) inhibits milk production efficiency. I think this research is very interesting and this conclusion is appropriate. However, I have other thoughts on the interpretation of the results that lead me to this conclusion. My interpretation about the data in Fig. 4-6 is that FUS inhibits mammary alveolar formation by affecting cell cycle in late pregnancy, resulting in fewer mammary follicles and low milk production during lactation. There are no histological data in late pregnancy in the current manuscript. I would strongly recommend examining how mammary alveolar formation is suppressed in late pregnancy (at least HE staining images about them). There are also some things that needed to be added or modified before acceptance.

Response: Thank you for your positive comments and suggestions. As suggested, we explored the effects of FUS on mammary alveolar formation in late pregnancy in the revised manuscript. At developmental stage P14 (pregnancy day 14), whole-mount analysis of mouse mammary tissue showed no significant differences in alveolar structure between the *Fus*-OE group and control groups, while notable atrophy and sparsity of alveolar structures were observed in *Fus*-OE mammary glands in late pregnancy (P16 and P18) (Fig. 4E). H&E staining further confirmed the sparse distribution of alveolar structures in the *Fus*-OE group in late pregnancy (Fig. 4F). We found that FUS expression began to be down-regulated in late pregnancy in wild-type mice (Fig. S1D), explaining the normal alveolar development in early pregnancy of

Fus-OE mice and abnormal development in late pregnancy.

Figures 4E, 4F, and S1D. (4E) Whole-mount carmine staining of mammary glands of *Fus*^{fl/fl}, WAP-Cre, and *Fus*-OE mice at indicated stages. Magnified areas are shown in black boxes. Scale bar: 1 mm. (4F) Representative H&E-stained images of mammary glands from *Fus*^{fl/fl}, WAP-Cre, and *Fus*-OE mice at P14 (n = 3 mice), P16 (n = 3 mice), P18 (n = 3 mice), L0 (n = 7 mice), L3 (n = 5 mice), L10 (n = 3 mice), and L20 (n = 3 mice). Scale bar: 200 μm (upper);

50 μ m (bottom). (S1D) Western blot analysis of FUS protein levels at various developmental stages (3-5 replicates per stage) in mouse mammary gland.

L44-L49

The lactation periods of mice, dog, humans, and cows are different even if the same eutherian species. Basically, the lactation period length is shorter for smaller animals and longer for larger ones. Therefore, when comparing lactation period lengths of marsupials and eutherians, species of animals of similar weight should be selected, respectively, in the manuscript.

Response: Thank you for your suggestion. We selected rats with a similar body weight to the sugar gliders to compare their lactation period lengths.

L66

Fill in the full name first, not the abbreviation. This is first of “FUS” in the manuscript.

Response: Thank you for your careful reading. We have added the full name of FUS in the revised manuscript.

L59-75

“Introduction” should refrain from describing excessive results.

Response: Thank you for your suggestion. We simplified the description of the results in the fourth paragraph of the Introduction section.

L84 (Fig.1)

To make valid comparison, the detailed information such as age, sexual cycle (If sugar gliders had one), days of pregnancy, and involution day after weaning should be added in in Method or Figure caption.

Response: Thank you for your suggestion. We have added detailed information on the age of the mice and sugar gliders, sexual cycles, stages of pregnancy, and involution days after weaning in the Figure legend.

L84 (Fig. 1B)

Fig. 1B is a mammary duct but not a mammary alveolus.

Response: Thank you for your suggestions. The corresponding descriptions were modified.

L87-89 (Fig. 1C)

It is difficult to distinguish between the mammary alveoli and ducts during lactation by HE staining. Especially the peripheral mammary ducts are about the same size as the mammary alveoli. To improve the paper, it is recommended to identify mammary ducts by immunostaining of breast duct marker molecules.

Response: Thank you for your suggestions. Accordingly, we selected CD133 as a marker to distinguish mammary alveoli and ducts based on previous reports¹⁵⁻¹⁷. Mammary ducts were clearly distinguished by CD133 immunofluorescence staining in mice and sugar gliders (Fig. S1A). The number of mammary gland ducts was recalculated and the new results (Fig. S1B) are similar to our previous findings (Fig. 1D) based on H&E staining images.

Figure S1A. Immunofluorescence staining for CD133 (red), EpCAM (green), and DAPI (blue) in mammary glands of sugar gliders and mice at virgin (V) and lactation stages (L). Scale bar: 50 μm; 20 μm. (S1B) Statistical analysis of number of CD133⁺ mammary gland ducts in sugar gliders and mice. Data are

means \pm SD. Unpaired t -test was used to evaluate statistical significance. $***$, $P < 0.0001$.

L102-108 (Fig.1 1H)

It would be good to show what milk component each peak indicates in graphs.

Response: Thank you for your suggestion. We have labeled the milk components represented by each peak in the graph (Fig. 1H).

Figure 1H. Raman spectroscopy and characteristic peaks of milk secretion and mammary gland contents in sugar gliders and mice during virgin and lactation stages. Raman shift: 1302 cm^{-1} (Amide III); 1440 cm^{-1} (CH₂ and CH₃ deformation vibrations); 1654 cm^{-1} (Amide I); 2850 cm^{-1} (stretching vibrations of CH, NH, and OH groups).

L266-272 (Fig.5C-G)

There is no causal relationship between dome formation and milk production. Dome formation occurs by changing in cell adhesion signaling. In this case, the expression levels of milk proteins in HC11 cells should be examined.

Response: Thank you for your suggestion. We agree that while dome formation assay is a recognized indicator for cell differentiation¹⁸, milk proteins also need to be examined. As suggested, we examined the expression of milk proteins and milk-related genes, including *Wap*, α -Casein, β -Casein, and *Prlr*. Results showed that these genes showed significant up-regulation after treatment with prolactin in the *Fus* knockdown groups, as indicated by RT-qPCR analysis (Fig. S5G). These findings are consistent

with previous research showing that HC11 cells can synthesize milk proteins when stimulated with lactogenic hormones, differentiate into alveolar-like cells, and form dome shaped structures^{19,20}. The size and number of domes are indicative of the degree of differentiation¹⁹. Taken together, these findings demonstrate that FUS can regulate milk production by inhibiting MEC differentiation.

Figure S5G. Relative mRNA expression of milk protein-related genes in HC11 cells after *Fus* knockdown, as determined by RT-qPCR. Data are means \pm SD. Unpaired *t*-test was used to evaluate statistical significance. **, $P < 0.01$; ***, $P < 0.001$; ****, $P < 0.0001$.

L208 (Fig. 3G)

Please explain in Discussion why the adverse effects of *Fus*-overexpression decrease in the second lactation than the first lactation.

Response: Thank you for your suggestion. We have included a discussion addressing why the adverse effects of *Fus* overexpression are less pronounced in the second lactation compared to the first at the end of the third paragraph in the Discussion section.

L233-246 (Fig. FE-H)

CK14 and WAP are used for *Fus* overexpression in the experiments. However, WAP is expressed in alveolar luminal epithelial cells whereas CK14 is expressed in basal cells/myoepithelial cells. Therefore, the cell types that are *Fus*-overexpressed are different. However, the results were the almost same in the mice using CK14 and WAP systems. Please explain about them in Discussion.

Response: Thank you for your suggestion. We have added a discussion to elaborate on why similar results were observed in mice using both the CK14 and WAP systems in the fourth paragraph of the Discussion section.

References

1. McLellan, M.A., N.A. Rosenthal, and A.R. Pinto. Cre-loxP-Mediated Recombination: General Principles and Experimental Considerations. *Current Protocols In Mouse Biology* **7** (2017).
2. Kos, C.H. Cre/loxP system for generating tissue-specific knockout mouse models. *Nutrition Reviews* **62**, 243-246 (2004).
3. Haller, M., et al. Ubiquitination and proteasomal degradation of ATG12 regulates its proapoptotic activity. *Autophagy* **10**, 2269-2278 (2014).
4. Trakatellis, A.C., A.E. Axelrod, and M. Montjar. ACTINOMYCIN D AND MESSENGER-RNA TURNOVER. *Nature* **203**, 1134-1136 (1964).
5. van den Heuvel, S. and N.J. Dyson. Conserved functions of the pRB and E2F families. *Nature Reviews. Molecular Cell Biology* **9**, 713-724 (2008).
6. Sherr, C.J. and J.M. Roberts. CDK inhibitors: positive and negative regulators of G1-phase progression. *Genes & Development* **13**, 1501-1512 (1999).
7. Blagosklonny, M.V. and A.B. Pardee. The restriction point of the cell cycle. *Cell Cycle (Georgetown, Tex.)* **1**, 103-110 (2002).
8. Ruijtenberg, S. and S. van den Heuvel. Coordinating cell proliferation and differentiation: Antagonism between cell cycle regulators and cell type-specific gene expression. *Cell Cycle (Georgetown, Tex.)* **15**, 196-212 (2016).
9. Stead, E., et al. Pluripotent cell division cycles are driven by ectopic Cdk2, cyclin A/E and E2F activities. *Oncogene* **21**, 8320-8333 (2002).
10. Lange, C. and F. Calegari. Cdks and cyclins link G1 length and differentiation of embryonic, neural and hematopoietic stem cells. *Cell Cycle (Georgetown, Tex.)* **9**, 1893-1900 (2010).
11. Sela, Y., et al. Human embryonic stem cells exhibit increased propensity to differentiate during the G1 phase prior to phosphorylation of retinoblastoma protein. *Stem Cells (Dayton, Ohio)* **30**, 1097-1108 (2012).
12. Calder, A., et al. Lengthened G1 phase indicates differentiation status in human embryonic stem cells. *Stem Cells and Development* **22**, 279-295 (2013).
13. Thoreen, C.C., et al. A unifying model for mTORC1-mediated regulation of mRNA translation. *Nature* **485**, 109-113 (2012).
14. Harvey, R.F., et al. Trans-acting translational regulatory RNA binding proteins. *Wiley Interdisciplinary Reviews. RNA* **9**, e1465 (2018).
15. dos Santos, C.O., et al. Molecular hierarchy of mammary differentiation yields refined markers of mammary stem cells. *Proceedings of the National Academy of Sciences of the United States of America* **110**, 7123-7130 (2013).
16. Dos Santos, C.O., et al. An epigenetic memory of pregnancy in the mouse mammary gland. *Cell Reports* **11**, 1102-1109 (2015).
17. Li, C.M.-C., et al. Aging-Associated Alterations in Mammary Epithelia and Stroma

- Revealed by Single-Cell RNA Sequencing. *Cell Reports* **33**, 108566 (2020).
18. Geletu, M., et al. Differentiation of Mouse Breast Epithelial HC11 and EpH4 Cells. *Journal of Visualized Experiments : JoVE* (2020).
 19. Morrison, B. and M.L. Cutler. Mouse Mammary Epithelial Cells form Mammospheres During Lactogenic Differentiation. *Journal of Visualized Experiments : JoVE* (2009).
 20. Wirl, G., et al. Mammary epithelial cell differentiation in vitro is regulated by an interplay of EGF action and tenascin-C downregulation. *Journal of Cell Science* **108 (Pt 6)**, 2445-2456 (1995).

REVIEWERS' COMMENTS

Reviewer #1 (Remarks to the Author):

The authors have taken into account most of my issues and offer now a lot more data to support the molecular understanding of FUS regulating milk production. I have no further comments.

Reviewer #2 (Remarks to the Author):

The authors have responded appropriately to my comments in revised manuscript.